# Unifying Counterfactual Data Augmentation and Architectural Inductive Biases in Offline Reinforcement Learning

## Abstract

Transformer-based models have recently achieved strong results in offline reinforcement learning by casting decision-making as sequence modeling. However, when trained purely on fixed datasets, they are prone to causal confusion: reliance on spurious correlations that predict reward in the data but do not reflect the true causal mechanisms of the environment. This issue is exacerbated by the weak inductive bias of Transformers, whose global attention is not aligned with the Markovian and causal dependencies of decision processes. We introduce the *Unified Causal Transformer (UCF)*, a framework that strengthens both the data and the model with causal consistency. On the data side, UCF employs a causal reward model to abduce exogenous factors and a counterfactual state generator to produce reward-preserving augmentations, yielding counterfactual trajectories that expose causal variability absent in observational data. On the model side, UCF integrates a causally structured hybrid architecture that combines separate modality-specific encoders for local dynamics with supervised attention for global reasoning, guiding the model to allocate representational capacity according to true causal dependencies. We evaluate UCF on two distinct sequential decision-making tasks—robotic control and recommendation—and demonstrate consistent gains in robustness and generalization over Transformer-based baselines. These results highlight the importance of causal consistency in both data and architecture for reliable offline policy learning.

## 1 Introduction

The success of high-capacity sequence models, particularly Transformers, in natural language processing (Devlin et al., 2019; Brown et al., 2020; Wolf et al., 2020) and computer vision (Chen et al., 2020; Ramesh et al., 2021; Reed et al., 2022) has inspired their use in decision-making. In offline reinforcement learning (RL), recent advances such as the Decision Transformer (DT) (Chen et al., 2021) demonstrate that sequential decision making can be cast as a sequence modeling problem. By processing trajectories as sequences of states, actions, and returns, these models predict future actions to achieve target objectives, leveraging pre-collected datasets without requiring additional environment interaction.

However, unlike text or images, decision-making tasks are governed by Markovian and causal dynamics. When trained purely on offline trajectories, Transformer-based models are prone to *causal confusion* (Lyle et al., 2021; Urpí et al., 2024): they may rely on spurious correlations that appear predictive in the dataset but do not reflect the true causal mechanisms of the environment (Gupta et al., 2023; Tien et al., 2022). For example, a model may incorrectly associate a background feature with high reward, leading to brittle policies that fail once that feature is absent at deployment. This limitation is compounded by the Transformer's highly flexible architecture. While its global self-attention mechanism is powerful for general sequence modeling, it lacks the specific inductive biases that reflect the underlying causal graph of a given decision process (Agarwal et al., 2023; Kim et al., 2024). Recent studies confirm that robustness and generalization in sequential decision-making depend critically on causal reasoning, showing that correlation-based learning alone is insufficient for reliable policy learning (Richens & Everitt, 2024). Addressing this causal misalignment is therefore

a central challenge for offline RL, where additional data collection is not possible to disambiguate spurious from causal associations.

We argue that addressing this challenge requires strengthening both the data and the model. On the data side, counterfactual augmentation offers a principled mechanism to expose causal variability that is missing from purely observational trajectories. By generating transitions under hypothetical interventions, the agent can learn to distinguish causal drivers of reward from spurious predictors (Lu et al., 2020; Chen et al., 2023). However, most existing augmentation methods assume access to environment interactions (Sun et al., 2024a; Cao et al., 2025), which is incompatible with the strictly offline setting we target. Moreover, some methods rely on strong structural assumptions, such as factored environments with independent entities that are not applicable to all environments (Pitis et al., 2022; Urpí et al., 2024). Our approach does not assume such structure, making it broadly applicable to standard continuous-control benchmarks such as Mujoco. On the model side, even with richer trajectories, a Transformer without appropriate inductive bias may still allocate attention according to superficial correlations (Hu et al., 2024; Kim et al., 2024). This motivates structuring the policy model to explicitly respect known causal dependencies, reducing the risk of overfitting to non-causal patterns. Together, causally consistent augmentation and causal structural guidance provide the two complementary ingredients needed for robust offline policy learning.

Concretely, we instantiate these two ingredients in a unified framework that we call the *Unified Causal Transformer (UCF)*. UCF integrates counterfactual data augmentation with a hybrid causal architecture for offline RL. First, a *causal reward model* (CRM) abduces latent exogenous factors from offline trajectories, enabling principled counterfactual reasoning. A *counterfactual state generator* (CSG) then proposes minimal modifications to observed states while preserving rewards under the abduced factors, yielding transitions that pass causal and reward consistency checks. Second, UCF employs a *causally structured hybrid architecture* that processes trajectories through modality-specific convolutions for local dynamics and a supervised attention layer for global reasoning. This design ensures that representational capacity is allocated to distinct causal roles, aligning predictions with the true decision-making structure. By jointly enriching both the data and the model with causal consistency, UCF provides a scalable and robust approach to offline policy learning. Our main contributions are as follows:

- We propose a novel offline RL framework that unifies counterfactual augmentation with causal architectural guidance.
- We introduce a causal reward model that infers exogenous factors and a counterfactual state generatorthat produces reward-preserving augmentations under explicit causal constraints.
- We develop a causally structured hybrid architecture that combines separate modality-specific encoders with supervised attention to respect causal dependencies in policy learning.
- We validate our framework on two distinct sequential decision-making tasks—robotic control and recommendation—showing consistent gains in generalization and robustness over correlation-based sequence models.

## 2 PRELIMINARIES

### 2.1 DECISION TRANSFORMER (DT)

We consider the offline reinforcement learning (RL) setting, where the agent has access only to a fixed dataset of trajectories and cannot interact with the environment. The environment is modeled as a Markov Decision Process (MDP), defined by $\mathcal{M} = (\mathcal{S}, \mathcal{A}, P, R, \gamma)$, where $\mathcal{S}$ is the state space, $\mathcal{A}$ is the action space, $P(s' \mid s, a)$ specifies the transition dynamics, $R(s, a)$ is the reward function, and $\gamma \in [0, 1)$ is the discount factor.

In offline RL, the dataset $\mathcal{D} = \{\tau_i\}_{i=1}^{N}$, with $\tau_i = (s_0, a_0, r_0, \ldots, s_T, a_T, r_T)$, is collected under one or more behavior policies, and the agent must learn a new policy without further environment interaction. The Decision Transformer (DT) (Chen et al., 2021) formulates offline RL as conditional sequence modeling. Instead of explicitly estimating value functions or dynamics, DT trains a Transformer autoregressively on trajectory data. Each trajectory is tokenized into a sequence:

$$\tau = (\hat{G}_0, s_0, a_0, \hat{G}_1, s_1, a_1, \ldots, \hat{G}_T, s_T, a_T),$$

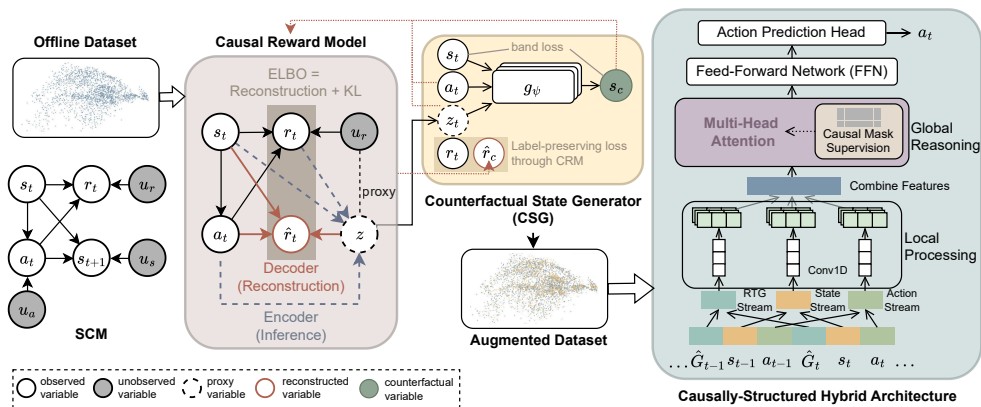

Figure 1: Overview of the proposed UCF framework. Left/Middle: A Causal Reward Model (CRM) and Counterfactual State Generator (CSG) jointly produce a causally consistent augmented dataset from offline trajectories. Right: A Causally-Structured Hybrid Architecture processes the data with local convolutions and global, causally supervised attention.

where $\hat{G}_t = \sum_{k=t}^{T} \gamma^{k-t} r_k$ denotes the returns-to-go (RTG). The model is trained to predict the action $a_t$ conditioned on a context window of the most recent $K$ tokens:

$$\pi_\theta(a_t \mid \hat{G}_{t-K+1:t}, s_{t-K+1:t}, a_{t-K+1:t-1}).$$

## 2.2 STRUCTURAL CAUSAL MODEL (SCM)

A Structural Causal Model (SCM) provides a formal framework for representing cause-effect relationships (Pearl, 2009). An SCM is a tuple $\mathcal{M}_c = (\mathcal{U}, \mathcal{V}, \mathcal{F}, P(\mathcal{U}))$, where $\mathcal{U}$ is a set of exogenous (external) variables, $\mathcal{V}$ is a set of endogenous (internal) variables, and $\mathcal{F}$ is a set of structural equations. Each equation $V_i := f_i(\text{Pa}(V_i), U_i)$ specifies how an endogenous variable $V_i \in \mathcal{V}$ is determined by its direct causes (parents) $\text{Pa}(V_i) \subseteq \mathcal{V}$ and an exogenous noise variable $U_i \in \mathcal{U}$, drawn from the distribution $P(\mathcal{U})$.

**Causal Relationships in MDPs**   The agent-environment interaction loop in an MDP can be formally described by an SCM (Peters et al., 2017; Zhang et al., 2020; Bennett et al., 2021; Shi et al., 2022). The causal mechanisms governing the policy, state transitions, and rewards are given by the following structural equations:

$$a_t := \pi(s_t, u_a), \quad s_{t+1} := f_s(s_t, a_t, u_s), \quad r_t := f_r(s_t, a_t, u_r), \tag{1}$$

where $a_t, s_{t+1}$, and $r_t$ are endogenous variables. The functions $\pi, f_s$, and $f_r$ represent the causal mechanisms for action selection, state transition, and reward generation, respectively. The terms $u_a, u_s$, and $u_r$ are mutually independent exogenous noise variables that account for the stochasticity in the system.

## 3 THE PROPOSED METHOD

We introduce a framework designed to improve offline RL by addressing causal consistency at both the data and model levels. The central idea is to enrich offline datasets with principled counterfactual transitions and to ensure the policy model processes information in a way that respects known causal structures. As illustrated in Figure 1, our framework is composed of two synergistic components: (i) a Counterfactual Transition Generation module that produces novel states which, under the original action and inferred unobserved context, would have yielded the identical reward, thereby creating label-preserving data augmentations; and (ii) a Causally-Structured Hybrid Architecture that integrates local dynamics modeling using separate modality-specific encoders with a global reasoning module. Crucially, this global module uses a final self-attention layer where specific heads are explicitly supervised to attend only to the direct causal parents of an action—namely, the current state

and the desired return-to-go. Together, these components enable a model to learn a more robust and generalizable policy from offline data.

### 3.1 COUNTERFACTUAL TRANSITION GENERATION

We aim to estimate the causal effect of hypothetical interventions on states using only offline data. We address the core counterfactual query: *What would the reward have been if the state had been different, while holding all else constant?* We follow Pearl's three steps of counterfactual reasoning (Pearl et al., 2000; Pearl, 2009): abduction (infer exogenous variables from data), action (surgically set the state to a hypothetical value), and prediction (evaluate the outcome under the modified world). To operationalize this, we adopt a two-stage process: (i) a *Causal Reward Model* (CRM) that captures the structural reward equation and provides the abduction step, and (ii) a *Counterfactual State Generator* (CSG) that edits states while holding the action and exogenous factors fixed, guided by the CRM and a controlled move band.

**Causal Reward Model (CRM)** We model the reward mechanism via the structural equation as defined in Equation (1):

$$r_t = f_r(s_t, a_t, u_r),$$

where $u_r$ are unobserved exogenous factors. We approximate this SCM using a Conditional Variational Autoencoder (CVAE), where the latent variable $z$ acts as a disturbance factor that captures variability in the reward mechanism. The encoder $q_\phi(z \mid s_t, a_t, r_t)$ performs abduction by inferring a posterior distribution over disturbance values that are consistent with the observed transition. The decoder $p_\theta(r_t \mid s_t, a_t, z)$ represents our learned structural function $\hat{f}_r$. The latent prior $p(z)$ serves as the prior distribution over disturbance factors. The model is trained by maximizing the ELBO:

$$\mathcal{L}_{\text{CVAE}} = \mathbb{E}_{q_\phi(z|s_t,a_t,r_t)}[\log p_\theta(r_t \mid s_t, a_t, z)] - \beta \, \text{KL}(q_\phi(z \mid s_t, a_t, r_t) \,\|\, p(z)). \quad (2)$$

After training, the latent variable $z$ is used as a disturbance factor conditioned on the observed transition. The encoder learns the posterior distribution $q_\phi(z \mid s_t, a_t, r_t)$, which captures the disturbance values that are consistent with the observed evidence. This follows the abduction step of counterfactual reasoning, where latent disturbances are inferred so that counterfactual predictions remain aligned with the reward mechanism. During counterfactual construction, the counterfactual state generator draws samples from this posterior to produce alternative states that satisfy reward consistency under the learned model.

**Counterfactual State Generator (CSG)** To generate a counterfactual for a given factual transition $(s_t, a_t, r_t)$, we first perform abduction using the CRM encoder by drawing a latent disturbance sample from the posterior, $z_t \sim q_\phi(z \mid s_t, a_t, r_t)$. This inferred context $z_t$ is then held constant and, along with the factual state and action, is used as input to the generator:

$$s_c = g_\psi(s_t, a_t, z_t).$$

The CSG is optimized to preserve the CRM reward under the factual action and fixed $z_t$, while keeping the edit size within a controlled band in *normalized* state space. Let $\sigma_s$ be the per-dimension standard deviation of states (computed on the dataset), and define the normalized move $\Delta_t = (s_c - s_t)/\sigma_s$. The loss is

$$\mathcal{L}_{\text{CSG}} = \left(\hat{f}_r(s_c, a_t, z_t) - r_t\right)^2 + \lambda_{\text{band}}\left(\left[\rho_{\text{low}} - \|\Delta_t\|_2\right]_+^2 + \left[\|\Delta_t\|_2 - \rho_{\text{high}}\right]_+^2\right), \quad (3)$$

where $[\cdot]_+ = \max(0, \cdot)$. The first term enforces *label preservation* under the SCM; the second term implements a *move band* that enforces state plausibility by discouraging trivial copies (too small moves) and off-manifold edits (too large moves). The details of the counterfacutal construction and move band are in Appendix H.

**Data Augmentation and Acceptance Gating** After the CRM and CSG are trained, we generate the augmented dataset. For each factual trajectory, we create a small number of augmented copies.

In each copy, we scan time steps that pass a reward quantile filter and propose $s_c = g_\psi(s_t, a_t, z_t)$. A candidate is only accepted if it passes two acceptance gates:

$$\left| \hat{f}_r(s_c, a_t, z_t) - r_t \right| \leq \varepsilon_r \quad \text{and} \quad \left\| \frac{s_c - s_t}{\sigma_s} \right\|_2 \leq \rho_{\text{high}}. \tag{4}$$

The tolerance $\epsilon_r$ is set adaptively. If accepted, we replace the observation token $s_t$ with the counterfactual $s_c$. The next observation $s_{t+1}$ is kept factual; we do not treat it as the successor of $s_c$. The final dataset is the union of all original trajectories and their edited copies. The details are in Appendix H.

**Validity of Counterfactual Augmentation**  Sampling $z_t$ from the posterior $q_\phi(z \mid s_t, a_t, r_t)$ applies the abduced disturbance during counterfactual generation, which satisfies the 'hold fixed' clause and preserves reward consistency. Conditioning $g_\psi$ on $(s_t, a_t, z_t)$ aligns edits with the action-conditioned reward mechanism $f_r(s, a, u_r)$. The band loss implements a minimal-change prior in normalized coordinates, which keeps augmented states near the data manifold.

**Proposition 1** (causal consistency of accepted counterfactuals). *Assume the learned CRM provides a sufficiently accurate approximation of the true reward SCM and the CSG is trained to (approximately) minimize equation 3. If a generated $s_c$ satisfies the acceptance gates $\left| \hat{f}_r(s_c, a_t, z_t) - r_t \right| \leq \varepsilon_r$ and $\|(s_c - s_t)/\sigma_s\|_2 \leq \rho_{\text{high}}$, then the augmented transition $(s_c, a_t, r_t)$ is a consistent sample from the post-intervention distribution implied by the reward SCM under the factual action and abduced exogenous variables. A detailed discussion of assumptions and proof is provided in Appendix C.2.*

## 3.2 Causally-Structured Hybrid Architecture

Enriching the dataset with causally-consistent transitions improves the quality of the input data; however, for the agent to fully benefit, the model architecture should also be designed to leverage this causal structure. To this end, we introduce a hybrid architecture that operates at both local and global levels. The model first encodes trajectory sequences using a stack of modality-specific convolutional layers, capturing local causal dynamics aligned with the Markov property. A final attention layer, guided by causal supervision, performs long-range reasoning to capture high-level dependencies relevant to goal-conditioned policy learning.

**Modeling Local Causal Dynamics with Separate Modality-Specific Encoders**  The convolutional backbone imposes a structural prior aligned with the local causal dynamics of MDPs. Trajectories are tokenized into sequences of return-to-go, state, and action tokens, and each modality is processed by its own 1D convolutional encoder. Separating the encoders by modality avoids representational interference and allows each signal type (state, action, and RTG) to be processed with an architecture suited to its structure. This preserves modality-specific information before the transformer performs global reasoning. The convolutional blocks then transform raw inputs into compact, temporally aware representations that serve as the local context for the transformer layers, while keeping the overall architecture simple and effective for sequence modeling.

**Supervised Attention for Global Reasoning**  After convolutional encoding, we apply a final multi-head self-attention block to perform global policy reasoning: determining which action $a_t$ best achieves the return-to-go $\hat{G}_t$ from the current state $s_t$. To align this reasoning with known causal dependencies, we supervise attention heads using explicit causal masks.

**Causal Attention Supervision**  We impose causal structure by supervising the attention weights for each action token to focus only on its direct parents: the state and return-to-go at the same timestep. This is encoded using a binary mask $M \in \{0, 1\}^{L \times L}$, where $M[i, j] = 1$ if token $j$ is a valid causal parent of token $i$. We construct a uniform target distribution: $q_{i,j} = \frac{1}{|S_i|} \cdot \mathbb{I}[j \in S_i]$, where $S_i = \{j : M[i, j] = 1\}$, and supervise the attention matrix $A$ using a cross-entropy loss:

$$\mathcal{L}_{\text{mask}} = \frac{1}{L} \sum_{i=1}^{L} \sum_{j \in S_i} -\frac{1}{|S_i|} \log(A_{i,j} + \epsilon), \tag{5}$$

with $\epsilon$ for numerical stability. The total objective becomes $\mathcal{L}_{\text{total}} = \mathcal{L}_{\text{action}} + \lambda \mathcal{L}_{\text{mask}}$, where $\lambda$ balances the causal constraint. This guides the model to respect causal structure while maintaining flexibility in learning global dependencies.

## 4 EXPERIMENT

We conduct a series of experiments to empirically validate the effectiveness of our proposed UCF. Our evaluation is designed to answer the following key research questions: (i): Does UCF outperform state-of-the-art offline RL baselines on standard offline RL benchmarks? (ii): Can UCF generalize beyond robotic control to other sequential decision-making tasks such as recommendation? (iii): What is the relative contribution of our two core components: the Counterfactual Transition Generation module and the Causally-Structured Hybrid Architecture? (iv): Does UCF demonstrate greater robustness than standard models when faced with causally misleading distractors in the environment?

### 4.1 LOCOMOTION AND ANTMAZE TASKS

In this section, we conduct a comprehensive empirical evaluation to answer our first research question. To this end, we test UCF on a diverse suite of locomotion and navigation tasks from the D4RL benchmark and compare its performance against leading value-based and Decision Transformer-based methods.

**Datasets** We evaluate UCF on a suite of continuous control tasks from the D4RL benchmark (Fu et al., 2020). Specifically, we consider tasks from two domains: locomotion and navigation. For locomotion, we use six datasets from three widely used agents—HalfCheetah, Hopper, and Walker2d—under three standard data regimes: medium (m), medium-replay (m-r), and medium-expert (m-e). For navigation, we include two datasets from the Antmaze environment: umaze (u) and umaze-diverse (u-d). This combined suite provides a comprehensive testbed for evaluating offline decision-making performance and generalization. Full details of the datasets are given in Appendix E.1.

**Baselines** We compare CaDM against several state-of-the-art offline RL algorithms. These include conservative offline RL methods: IQL (Kostrikov et al., 2021) and CQL (Kumar et al., 2020). We also benchmark against DT-based methods: the standard DT (Chen et al., 2021), DC (Kim et al., 2024), and LSDT(Wang et al., 2025a). Full details of the baselines are given in Appendix E.1.

**Overall Results** The performance of our method, UCF, alongside state-of-the-art baselines is presented in Table 1. All scores are expert-normalized returns as specified by the D4RL benchmark (Fu et al., 2020). The results clearly demonstrate that UCF achieves state-of-the-art or highly competitive performance across the full suite of MuJoCo locomotion and Antmaze navigation tasks. On the MuJoCo locomotion tasks, UCF stands out as the top-performing method in 6 out of the 9 environments. It shows particularly dominant results in the Hopper and Walker2d domains and consistently excels on the challenging medium-expert and medium-replay datasets. This indicates a strong ability to leverage mixed-quality data, a common challenge in offline reinforcement learning. Furthermore, UCF proves its versatility on the Antmaze tasks, which are characterized by sparse rewards and require long-horizon, goal-conditioned reasoning. In both the umaze and umaze-d environments, UCF achieves highly competitive scores that are within 5% of the best-performing methods (IQL and CQL, respectively). This strong performance in both dense-reward locomotion and sparse-reward navigation highlights UCF's effectiveness as a general-purpose algorithm. Overall, the empirical results confirm that UCF is a robust and powerful method for offline decision-making, capable of matching or exceeding the performance of leading value-based and transformer-based approaches across a diverse set of challenges.

### 4.2 RECOMMENDATION TASKS

**Datasets** For the recommendation domain, we evaluate UCF on three large-scale, real-world datasets: KuaiRand (Gao et al., 2022b), KuaiRec (Gao et al., 2022a), and VirtualTB (Shi et al., 2019). Full details of the datasets are given in Appendix E.2.

Table 1: Offline results on the MuJoCo and Antmaze datasets. We report the expert-normalized returns, averaged across 5 random seeds for MuJoCo and 4 for Antmaze. The boldface numbers denote the best score and any scores within 5% of the best for each dataset.

| Method | H-Cheetah | | | Hopper | | | Walker2d | | | Antmaze | |
|---|---|---|---|---|---|---|---|---|---|---|---|
| | -m | -m-r | -m-e | -m | -m-r | -m-e | -m | -m-r | -m-e | -umaze | -umaze-d |
| IQL | **47.4** | **44.2** | 86.7 | 66.3 | **94.7** | 91.5 | 78.3 | 73.9 | **109.6** | **87.5** | 62.2 |
| CQL | 44.0 | 37.5 | **91.6** | 58.5 | **95.0** | 105.4 | 72.5 | **77.2** | 108.8 | 74.0 | **84.0** |
| DT | 42.6 | 36.6 | 86.8 | 67.6 | 82.7 | **107.6** | 74.0 | 66.6 | **108.1** | 69.8 | 70.3 |
| DC | 43.0 | 41.3 | **93.0** | 92.6 | 94.2 | 110.4 | 79.2 | 76.6 | **109.6** | 82.2 | 78.5 |
| LSDT | 43.6 | **42.9** | 93.2 | 87.2 | 93.9 | 111.7 | 81.0 | 74.7 | 109.8 | 80.0 | **83.2** |
| **UCF** | 44.9 | 43.5 | 93.1 | 93.4 | 94.8 | 112.0 | 82.5 | 77.0 | 110.7 | 83.1 | 84.5 |

**Baselines**    In this domain, we compare UCF against recent DT-based baselines tailored for recommender systems: DT4Rec (Zhao et al., 2023), CDT4Rec (Wang et al., 2023), and EDT4Rec (Chen et al., 2024). Full details of the baselines are given in Appendix E.2.

**Overall Results**    The evaluation results on the three recommendation datasets are presented in Table 2. The findings are unequivocal: UCF consistently outperforms all specialized, state-of-the-art Decision Transformer baselines across every metric on all datasets. Specifically, UCF achieves the highest Cumulative Reward ($\mathcal{R}_{cum}$) and Average Reward ($\mathcal{R}_{avg}$) on KuaiRand, KuaiRec, and VirtualTB, often by a clear margin over the second-best methods. This strong performance provides an affirmative answer to our second research question (RQ ii), demonstrating that UCF generalize effectively beyond robotic control to the distinct domain of sequential recommendation. The ability of our general-purpose model to surpass domain-specialized methods highlights the robustness and broad applicability of our proposed architecture.

Table 2: Evaluation results on recommendation datasets. Metrics include Cumulative Reward ($\mathcal{R}_{cum}$) and Average Reward ($\mathcal{R}_{avg}$). **Bold** indicates the best performance per metric, and * marks the second-best.

| Method | KuaiRand | | KuaiRec | | VirtualTB | |
|---|---|---|---|---|---|---|
| | $\mathcal{R}_{cum}$ | $\mathcal{R}_{avg}$ | $\mathcal{R}_{cum}$ | $\mathcal{R}_{avg}$ | $\mathcal{R}_{cum}$ | $\mathcal{R}_{avg}$ |
| DT4Rec | $6.8172 \pm 2.45$ | $0.5686 \pm 0.21$ | $28.5418 \pm 10.42$ | $0.8798 \pm 0.34$ | $76.7871 \pm 22.63$ | $5.4420 \pm 1.72$ |
| CDT4Rec | $7.3271 \pm 1.98$ | $0.6508^* \pm 0.19$ | $30.4888 \pm 10.19$ | $1.0061 \pm 0.35$ | $79.2101 \pm 22.38$ | $5.6490 \pm 1.61$ |
| EDT4Rec | $7.5817^* \pm 1.84$ | $0.6497 \pm 0.17$ | $31.0726^* \pm 10.98$ | $1.0397^* \pm 0.39$ | $79.6651^* \pm 21.67$ | $5.6741^* \pm 1.54$ |
| UCF | $\mathbf{7.6221} \pm 1.79$ | $\mathbf{0.6554} \pm 0.16$ | $\mathbf{31.8721} \pm 10.55$ | $\mathbf{1.0582} \pm 0.40$ | $\mathbf{80.4241} \pm 21.15$ | $\mathbf{5.6957} \pm 1.51$ |

## 4.3    ABLATION STUDY

**Contributions of Two Core Components**    To investigate the contributions of our two core components, we evaluate two variants of our model: w/o Count, which removes counterfactual data augmentation, and w/o Arch, which replaces our causal architecture with a standard DT backbone. The results in Table 3 highlight their distinct yet complementary roles. On the dense-reward MuJoCo tasks, the causal hybrid architecture is the dominant contributor. Removing it (w/o Arch) causes a clear performance drop, especially on the more diverse medium-replay datasets for Hopper and Walker2d. In contrast, removing data augmentation (w/o Count) leads to only small degradations, suggesting that the architecture can already extract strong policies when rewards are plentiful and coverage is reasonable. On the sparse-reward AntMaze tasks, augmentation becomes more important: w/o Count consistently underperforms UCF, while w/o Arch suffers an even larger drop, showing that both components are needed. These results confirm our hypothesis: counterfactual augmentation enriches the dataset, while the causal hybrid architecture is essential for leveraging this variability to produce robust long-horizon plans.

Table 3: Ablation study of UCF components on MuJoCo and Antmaze tasks. We compare the full model against variants without the Counterfactual Transition Generation (w/o Count) and without the Causally-Structured Hybrid Architecture (w/o Arch). Bold numbers denote the best score and any scores within 5% of the best for each dataset.

| Method | H-Cheetah | | | Hopper | | | Walker2d | | | Antmaze | |
|---|---|---|---|---|---|---|---|---|---|---|---|
| | -m | -m-r | -m-e | -m | -m-r | -m-e | -m | -m-r | -m-e | -umaze | -umaze-d |
| **UCF** | **44.9** | **43.5** | **93.1** | **93.4** | **94.8** | **112.0** | **82.5** | **77.0** | **110.7** | **83.1** | **84.5** |
| w/o Count | **43.8** | **42.7** | **93.0** | **92.8** | **94.3** | **111.2** | **81.5** | **76.8** | **110.0** | **81.5** | **82.5** |
| w/o Arch | **43.2** | 41.0 | **91.0** | 84.2 | **91.5** | **110.0** | **79.8** | 73.5 | **109.2** | 78.0 | 79.5 |
| DT | 42.6 | 36.6 | 86.8 | 67.6 | 82.7 | 107.6 | 74.0 | 66.6 | 108.1 | 69.8 | 70.3 |
| DC | **43.0** | 41.3 | **93.0** | **92.6** | **94.2** | **110.4** | 79.2 | **76.6** | **109.6** | **82.2** | 78.5 |
| LSDT | **43.6** | **42.9** | **93.2** | 87.2 | **93.9** | **111.7** | **81.0** | 74.7 | **109.8** | 80.0 | **83.2** |

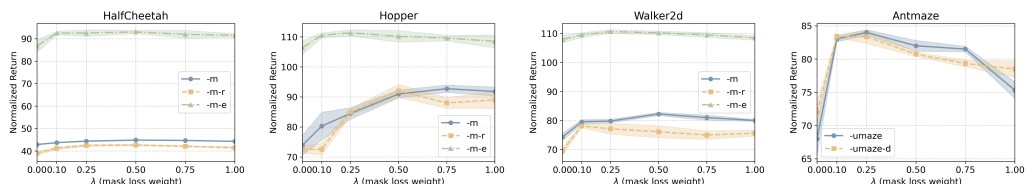

Figure 2: Ablation on the causal supervision weight $\lambda$ across MuJoCo locomotion tasks and Antmaze. Each plot reports normalized returns (mean $\pm$ standard deviation over 5 seeds).

**Effect of Causal Supervision Weight $\lambda$.** We investigate the effect of the causal supervision weight $\lambda$, which balances the action prediction loss and the mask supervision loss. Setting $\lambda = 0$ removes causal supervision entirely, reducing UCF to a standard Transformer; $\lambda = 1$ enforces full adherence to the causal mask. Figure 2 shows that moderate values of $\lambda$ typically yield the best trade-off. In simple environments such as Hopper, larger $\lambda$ improves stability and performance. In more complex domains such as HalfCheetah or Antmaze, too much supervision can constrain the model, and intermediate $\lambda$ values achieve the strongest results. This confirms that causal supervision is beneficial, but its strength should adapt to task complexity.

### 4.4 Robustness to Spurious Correlations

To test whether UCF mitigates causal confusion, we construct semi-synthetic datasets by injecting a binary distractor feature into standard D4RL benchmarks (HalfCheetah-medium, Hopper-medium, and Walker2d-medium). The distractor is correlated with reward in the training set but has no causal effect on the environment. At test time, we intervene by fixing the distractor to 0.0, breaking the correlation. Figure 3 reports normalized returns (mean $\pm$ standard deviation across 5 seeds) under this intervention. DT suffers a severe collapse in performance, confirming that it overfit to the spurious distractor. DC shows moderate robustness, but its reliance on correlation still leads to a significant drop. In contrast, UCF maintains strong performance, with only a minor degradation compared to training on the original dataset. These results provide direct evidence that UCF's design—combining counterfactual data augmentation and causally supervised attention—successfully prevents the model from treating the distractor as causal. Full dataset construction and evaluation details are in Appendix G.4.

## 5 Related Work

### 5.1 Transformer Architectures for Reinforcement Learning

Recent work has reframed sequential decision making as a sequence modeling problem, where trajectories of returns-to-go, states, and actions are treated as tokens processed by Transformer-based architectures. DT (Chen et al., 2021) demonstrated that conditioning on return-to-go enables policy

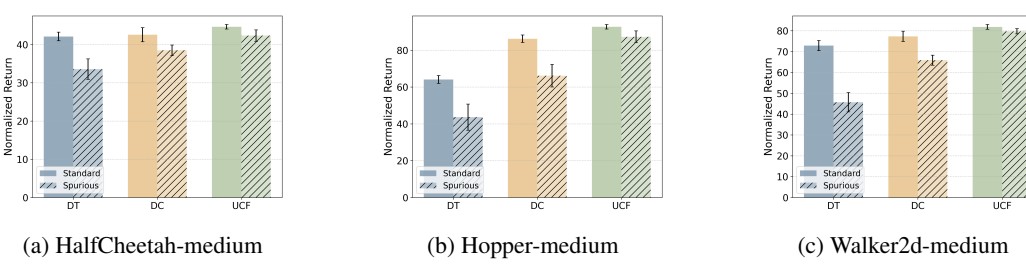

(a) HalfCheetah-medium      (b) Hopper-medium      (c) Walker2d-medium

Figure 3: Robustness to spurious distractors. Expert-normalized returns for DT, DC, and UCF when trained on datasets with a spurious binary feature and evaluated under intervention.

learning without value functions or explicit dynamics models. Trajectory Transformer (Janner et al., 2021) extended this perspective by discretizing trajectories and using beam search for planning. Beyond purely autoregressive modeling, masked prediction has been explored as an alternative training paradigm. Uni[MASK] (Carroll et al., 2022) proposed a unified framework where diverse inference tasks are cast as different masking patterns, while MaskDP (Liu et al., 2022) applied masked autoencoding to state–action trajectories, showing strong zero-shot and fine-tuning performance. At the architectural level, several works have explored combining attention with convolution to better align with the structure of decision-making problems. DC (Kim et al., 2024) replaced most attention blocks with lightweight convolutional mixers, showing that local filtering is often sufficient for Markovian dynamics while retaining a final attention block for long-range reasoning. LSDT (Wang et al., 2025a) proposed a parallel hybrid design where each block splits into a convolution branch for local dependencies and an attention branch for global dependencies, with channel ratios controlling their relative contributions. This sequence modeling paradigm has also been extended to recommendation tasks. CDT4Rec (Wang et al., 2023) incorporated a causal reward estimator, EDT4Rec (Chen et al., 2024) leveraged entropy regularization and reward relabeling for learning from suboptimal data, and MaskRDT (Wang et al., 2025b) improved efficiency on long user histories through retentive networks and adaptive masking. Our approach UCF, differs from prior architectures by its explicit causal grounding. Unlike DC and LSDT, which combine convolution and attention without causal guidance, UCF augments offline data with counterfactual trajectories that preserve causal consistency and constrains attention to focus on true causal parents. This integration of causal augmentation and causal priors yields policies that are more robust and generalizable.

## 5.2 CAUSAL REINFORCEMENT LEARNING

A complementary line of research introduces causality into RL to address spurious correlations and improve generalization (Zeng et al., 2025). CDL (Wang et al., 2022) learns task-independent state abstractions by uncovering the causal structure of environment dynamics. ACE (**?**) proposes causality-aware entropy regularization that weights action dimensions by their causal influence on rewards, improving exploration efficiency in continuous control. CSR (Yang et al., 2025) develops causality-guided self-adaptive representations that detect distribution shifts, expand causal graphs to accommodate new variables, and prune irrelevant factors, enabling more generalizable policy transfer. A particularly active direction focuses on counterfactual reasoning. MOCODA (Pitis et al., 2022) generates counterfactual transitions using a factored dynamics model for improved out-of-distribution generalization. CAIAC (Urpí et al., 2024) targets robotic manipulation, augmenting data by swapping action-independent factors between trajectories. ACAMDA (Sun et al., 2024b) applies adversarial counterfactual augmentation to enforce causally consistent dynamics. More recently, CIP (Cao et al., 2025) combines counterfactual data augmentation with causality-aware empowerment to improve sample efficiency across domains. Unlike prior counterfactual augmentation methods, which either assume factored environments with independent entities (MOCODA, CAIAC) or rely on online interaction with the environment (ACAMDA, CIP), our method is explicitly tailored for strictly offline RL where only a fixed dataset is available. Crucially, we do not assume a factorized state space, making our approach applicable to standard continuous-control benchmarks such as Mujoco. On the model side, we further introduce a causally structured hybrid architecture with supervised attention, ensuring that the agent can fully benefit from counterfactual augmentation by aligning its reasoning with true causal dependencies rather than spurious correlations.

## 6 CONCLUSION

We presented the Unified Causal Transformer (UCF), a framework that strengthens offline reinforcement learning through causal consistency at both the data and model levels. On the data side, UCF introduces counterfactual augmentation by combining a causal reward model with a counterfactual state generator, producing reward-preserving transitions that expand the training set without requiring online interaction or factorized environments. On the model side, UCF employs a causally structured hybrid architecture that integrates convolutional modeling of local dynamics with supervised attention for global reasoning, ensuring that predictions align with true causal parents. Experiments on robotic control and recommendation tasks demonstrate that UCF achieves improved robustness, generalization, and resistance to spurious correlations compared to existing baselines.

While UCF advances causal consistency in offline RL, several limitations remain. First, the quality of counterfactual augmentation depends on the accuracy of the learned causal reward model; inaccuracies in modeling complex reward mechanisms may reduce the validity of generated samples. Second, although our causal supervision improves interpretability and robustness, it introduces additional hyperparameters (e.g., supervision strength, window size) that require careful tuning. Finally, UCF has so far been evaluated on standard continuous-control and recommendation benchmarks; its effectiveness in large-scale, high-dimensional environments with richer structure remains an open question. Future work will explore integrating stronger causal discovery techniques into the augmentation process, extending UCF to partially observable or multi-agent settings, and studying how causal supervision can be dynamically adapted during training to further improve scalability and generalization.

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

## A    THE USAGE OF LLM

In preparing this manuscript, we employed a large language model (LLM) strictly as a writing assistant. Its role was limited to grammar checking, stylistic polishing, and improving readability of the text. All technical content, experimental design, and results were produced by the authors.

## B    BACKGROUND ON CAUSAL INFERENCE

Here, we introduce fundamental causal modeling concepts (Pearl, 2009; Peters et al., 2017) that underpin our methodology and theoretical analysis.

**Structural Causal Models**    Structural Causal Models (SCMs) formalize the data-generating process by specifying how each variable is causally determined by its parents and exogenous noise. An SCM is associated with a directed acyclic graph (DAG) that encodes causal relationships among variables.

**Definition B.1** (Structural Causal Model (Pearl, 2009)). *A Structural Causal Model (SCM)* $\mathcal{M} = (\mathcal{G}, \mathbf{S}, P_{\mathbf{U}})$ *consists of:*

- *A directed acyclic graph (DAG) $\mathcal{G} = (\mathbf{V}, \mathcal{E})$, where $\mathbf{V}$ is a set of endogenous variables and $\mathcal{E}$ is the set of directed edges representing direct causal relationships;*

- *A collection of structural assignments $\mathbf{S} = \{X_i = f_i(\mathrm{PA}_i, U_i)\}$ for each $X_i \in \mathbf{V}$, where $\mathrm{PA}_i \subseteq \mathbf{V} \setminus \{X_i\}$ are the parent variables of $X_i$ in $\mathcal{G}$, and $U_i \in \mathbf{U}$ are exogenous noise variables;*

- *A joint distribution $P_{\mathbf{U}}$ over the exogenous variables $\mathbf{U} = \{U_1, \ldots, U_n\}$, typically assumed to be mutually independent.*

*An SCM $\mathcal{M}$ induces a joint observational distribution over $\mathbf{V}$ according to the structural assignments and exogenous distribution.*

**Definition B.2** (Intervention (Pearl, 2009))**.** *An* intervention *in an SCM $\mathcal{M} = (\mathcal{G}, \mathbf{S}, P_{\mathbf{U}})$ corresponds to replacing the structural assignment for a variable $X_j \in \mathbf{V}$ with a new mechanism:*

$$X_j = \hat{f}_j(\widehat{\mathrm{PA}}_j, \hat{U}_j),$$

*resulting in a modified model $\hat{\mathcal{M}}$. The new model $\hat{\mathcal{M}}$ induces a different distribution over the variables $\mathbf{V}$, referred to as the* interventional distribution*:*

$$P_{\hat{\mathcal{M}}}(\mathbf{V}) = P_{\mathcal{M}}(\mathbf{V} \mid \mathrm{do}(X_j = \hat{f}_j(\widehat{\mathrm{PA}}_j, \hat{U}_j))).$$

**Definition B.3** (Causal Effect Identifiability (Pearl, 2009))**.** *The causal effect of $X$ on $Y$ is* identifiable *from a graph $\mathcal{G}$ if the quantity $P(y \mid \mathrm{do}(x))$ can be computed uniquely from any positive probability of the observed variables. That is, if*

$$P_{M_1}(y \mid \mathrm{do}(x)) = P_{M_2}(y \mid \mathrm{do}(x))$$

*for every pair of models $M_1$ and $M_2$ such that $P_{M_1}(\nu) = P_{M_2}(\nu) > 0$ and $\mathcal{G}(M_1) = \mathcal{G}(M_2) = \mathcal{G}$.*

**Theorem B.1** (Three Steps of Counterfactual Reasoning (Pearl, 2009))**.** *Computing a counterfactual requires a three-step process:*

1. ***Abduction:*** *Condition the distribution of the exogenous variables $u_r$ on the observed evidence $e$, obtaining $P(u_r \mid e)$.*

2. ***Action:*** *Modify the SCM by performing a surgical intervention using the do()-operator, forcing variables to take hypothetical values (e.g., setting the state to $s_c$).*

3. ***Prediction:*** *Use the modified model together with $P(u_r \mid e)$ to compute the counterfactual outcome.*

## C  ASSUMPTIONS AND PROPOSITIONS

### C.1  ASSUMPTIONS

**A1 (Markov property) (Pearl, 2009).** The environment is Markovian. The joint over $\mathcal{V} = \{s_t, a_t, r_t, s_{t+1}\}$ factorizes as $p(s_t)\, p(a_t \mid s_t)\, p(r_t \mid s_t, a_t)\, p(s_{t+1} \mid s_t, a_t)$. *Use:* underlies the SCM in Eq. (1), CRM training, and the counterfactual edit that changes only $s_t$.

**A2 (Faithfulness) (Pearl, 2009).** The observed distribution is faithful to the causal DAG $\mathcal{G}$: conditional independences coincide with $d$-separations (no cancellation). *Use:* justifies that the supervised attention mask encodes true parent sets of $a_t$.

**A3 (Causal minimality) (Pearl, 2009).** The DAG has no redundant edges: for every edge $X \to Y$, $X \not\perp\!\!\!\perp Y \mid \mathrm{Pa}(Y) \setminus \{X\}$. *Use:* prevents spurious parents in the mask and supports interpretability.

**A4 (Temporal causality).** No backward or instantaneous cycles; edges respect time order. *Use:* supports abduction at time $t$ and interventions on $s_t$ while holding $a_t$ and exogenous variables fixed.

### C.2  PROPOSITION ON CAUSAL CONSISTENCY

We restate Proposition 1 in an assumption–guarantee form. The argument follows Pearl's abduction–action–prediction framework.

**Additional conditions.**

1. **Approximate reward model.** The learned Causal Reward Model (CRM) approximates the true reward mechanism with error at most $\delta$, i.e.,
$$\left|\hat{f}_r(s, a, z) - f_r(s, a, u_r)\right| \leq \delta,$$
whenever $z$ is produced by the CRM encoder from $(s, a, r)$.

2. **Generator training.** The Counterfactual State Generator (CSG) is trained with the objective in Eq. equation 3 and proposes edits within an upper move bound $\rho_{\text{high}}$.

3. **Acceptance test.** A proposed counterfactual state $s_c$ is accepted only if it satisfies
$$\left|\hat{f}_r(s_c, a_t, z_t) - r_t\right| \leq \varepsilon_r \quad \text{and} \quad \|(s_c - s_t)/\sigma_s\|_2 \leq \rho_{\text{high}}.$$

**Guarantee.** Under Assumptions A1–A4 and the above conditions, every accepted counterfactual transition $(s_c, a_t, r_t)$ corresponds to an approximate sample from the interventional distribution
$$P(r \mid do(s_t \leftarrow s_c), a_t, u_r),$$
with approximation error bounded by $\delta + \varepsilon_r$.

**Argument.** The reward SCM is defined as $r = f_r(s, a, u_r)$ with exogenous noise $u_r \sim P(u_r)$ independent of $(s, a)$.

*Abduction.* Given $(s_t, a_t, r_t)$, we infer a posterior over $u_r$; in practice, the CRM encoder provides $z_t$ as a proxy for the abduced $u_r$.

*Action.* A counterfactual state $s_c = g_\psi(s_t, a_t, z_t)$ is generated, corresponding to the intervention $do(s_t \leftarrow s_c)$ while keeping $(a_t, u_r)$ fixed.

*Prediction.* The CRM decoder evaluates $\hat{f}_r(s_c, a_t, z_t)$. By the approximation property, $\hat{f}_r(s_c, a_t, z_t) \approx f_r(s_c, a_t, u_r)$ within $\delta$.

*Acceptance.* If $\hat{f}_r(s_c, a_t, z_t)$ is within $\varepsilon_r$ of $r_t = f_r(s_t, a_t, u_r)$, then
$$\left|f_r(s_c, a_t, u_r) - f_r(s_t, a_t, u_r)\right| \leq \delta + \varepsilon_r.$$
Thus the counterfactual reward matches the factual reward up to tolerance, while the move constraint ensures $s_c$ remains near the support of the data. Therefore, $(s_c, a_t, r_t)$ is an approximate sample from the interventional distribution under the factual action and abduced exogenous variables.

# D EXTENDED RELATED WORK

## D.1 CAUSALITY IN ATTENTION MECHANISMS

Recent work has explored integrating causality into attention to enhance interpretability and generalization. Some studies interpret attention through a causal lens, such as Rohekar et al. (2023), who treat self-attention as estimating an SCM via constraint-based methods. Others embed attention into causal frameworks, like CAL (Sui et al., 2022) and CAL+ (Sui et al., 2024), which use attention to identify causal features in GNNs. Intervention-based designs have also emerged, e.g., Ge et al. (2023) introduce Social Cross Attention with learnable variables representing confounder strata to deconfound human trajectory prediction. In contrast, our approach directly supervises attention heads using a predefined causal graph, aligning attention patterns with known structural dependencies in sequential decision-making tasks.

# E DATASETS AND BASELINES

## E.1 ROBOTIC CONTROL TASK

**Datasets** For robotic control experiments, we use datasets from the D4RL benchmark (Fu et al., 2020), which are widely used in offline reinforcement learning research. These datasets are generated using the MuJoCo physics simulator and consist of pre-collected trajectories from various environments and policy qualities. Specifically, we evaluate on:

- `HalfCheetah`: A 2D bipedal cheetah-like robot aiming to run.
- `Hopper`: A 2D one-legged hopper robot aiming to hop forward.
- `Walker2d`: A 2D bipedal robot aiming to walk.

For each of these environments, we use three dataset types reflecting different data quality and collection strategies:

- `-medium (m)`: Trajectories collected by a policy trained to a medium level of performance and then rolled out.
- `-medium-replay (m-r)`: The full replay buffer contents of an agent trained to a medium level of performance.
- `-medium-expert (m-e)`: A 50/50 mix of trajectories from a medium policy and an expert policy.

**Baselines** We compare CaDM against several state-of-the-art and representative offline RL algorithms in the robotic control domain.

- **Offline RL Methods:**
  - `IQL (Implicit Q-Learning) (Kostrikov et al., 2021)`: An offline Q-learning method that learns Q-functions by implicitly defining them via expectile regression, avoiding explicit policy constraints or out-of-distribution action queries.
  - `CQL (Conservative Q-Learning) (Kumar et al., 2020)`: A widely used offline RL algorithm that learns a conservative Q-function by adding a regularization term to the standard Bellman error. This term penalizes high Q-values for actions outside the dataset distribution and encourages low Q-values for them, mitigating overestimation issues.
- **Decision Transformer (DT) and Variants:**
  - `DT (Decision Transformer) (Chen et al., 2021)`: The standard Decision Transformer that models RL as a sequence modeling problem, predicting actions autoregressively based on desired returns-to-go, past states, and actions.
  - `DC (Decision ConvFormer) (Kim et al., 2024)`: A DT variant that integrates convolutional layers, potentially to better capture local features or spatial relationships in states, which can be beneficial in certain control tasks.
  - `LSDT (Long-Short Decision Transformer) (Wang et al., 2025a)`: A Decision Transformer variant that augments the model with a long–short temporal module, enabling it to capture both short-term transitions and long-range dependencies more effectively.

E.2 RECOMMENDATION TASK

**Datasets** For evaluating CaDM on recommendation tasks, we selected the following publicly available datasets, known for their scale and real-world relevance:

- `KuaiRand`: An unbiased sequential recommendation dataset collected from the recommendation logs of the Kuaishou video-sharing mobile app (Gao et al., 2022b). It is notable for including millions of intervened interactions where items were randomly exposed within standard recommendation feeds, which helps in studying and mitigating exposure bias. It provides rich side information, including user IDs, interaction timestamps, and features for users and items, across various collection policies.
- `KuaiRec`: Another dataset from Kuaishou, KuaiRec is distinguished by its "fully-observed" user-item interaction matrix for a subset of users and items, meaning nearly all preferences are known (Gao et al., 2022a). This dense interaction data (e.g., 1,411 users and 3,327 items with 99.6% density in its "small matrix") is valuable for evaluating recommendation models without suffering severely from missing data issues, and for research in unbiased recommendation, interactive RL, and off-policy evaluation. It also contains a larger, sparser "big matrix" and side information like item categories and a social network.

- `VirtualTB (Virtual Taobao)`: An online simulation platform that mimics a real-world e-commerce environment (Taobao) for developing and testing recommender systems (Shi et al., 2019). It is trained on hundreds of millions of real user data points and generates virtual customers with dynamic and static features. VirtualTB allows RL agents to interact with the simulated environment, receive feedback (e.g., clicks), and be evaluated on metrics like Click-Through Rate (CTR).

**Baselines** For the recommendation tasks, we compare CaDM with the following state-of-the-art Decision Transformer-based models designed for recommender systems:

- `CDT4Rec (Causal Decision Transformer for Recommender Systems) (Wang et al., 2023)`: This model adapts the Decision Transformer framework for recommendation by incorporating a causal mechanism. It aims to address the challenge of reward function design by estimating rewards based on the causal relationships inferred from user behavior within the transformer architecture.

- `DT4Rec (Decision Transformer for Recommender Systems) (Zhao et al., 2023)`: This approach applies the Decision Transformer to focus on user retention in recommender systems . It often employs specific reward prompting strategies tailored for recommendation scenarios to guide the DT model.

- `EDT4Rec (Max-Entropy enhanced Decision Transformer with Reward Relabeling for Offline RLRS) (Chen et al., 2024)`: This model enhances Decision Transformer-based methods for recommendation by tackling limitations such as "stitching" suboptimal trajectories and insufficient online exploration. It integrates max-entropy regularization to encourage exploration and a reward relabeling technique (often based on learned Q-values from methods like CQL) to improve learning from suboptimal data.

# F  ALGORITHMS FOR UCF

---

**Algorithm 1** Unified Causal Transformer (UCF): End-to-end training on augmented data

---

**Require:** Offline dataset $\mathcal{D}$; CRM encoder $q_\phi$ and decoder $p_\theta$; CSG $g_\psi$; UCF policy model $\pi_\omega$ (hybrid conv + final attention); mask weight $\lambda$; augmentation budget $K$ per trajectory; thresholds $(\varepsilon_r, \rho_{\text{high}})$; state normalization $\sigma_s$; optimizers $\text{Opt}_\phi, \text{Opt}_\theta, \text{Opt}_\psi, \text{Opt}_\omega$

1: **Train CRM** $(q_\phi, p_\theta)$ on $\mathcal{D}$ by maximizing the ELBO (Alg. 2)
2: **Train CSG** $g_\psi$ with banded move and reward-preservation (Alg. 3)
3: **Augment** $\mathcal{D}$ using $(q_\phi, p_\theta, g_\psi)$ and the acceptance gate (Alg. 5, 4); get $\mathcal{D}_{\text{aug}}$
4: **Train UCF policy** $\pi_\omega$ on $\mathcal{D}_{\text{aug}}$ with action loss + mask loss (Alg. 6)
5: **return** trained policy $\pi_\omega$

---

---

**Algorithm 2** Training the Causal Reward Model (CRM) as a CVAE

---

**Require:** Offline dataset $\mathcal{D} = \{(s_t, a_t, r_t)\}$; prior $p(z)$ (e.g., $\mathcal{N}(0, I)$); encoder $q_\phi(z \mid s, a, r)$; decoder $p_\theta(r \mid s, a, z)$; ELBO weight $\beta$; optimizer Opt
1: Initialize parameters $\phi, \theta$
2: **while** not converged **do**
3:      Sample minibatch $\{(s_t, a_t, r_t)\}_{t=1}^{B} \sim \mathcal{D}$
4:      Encode: $(\mu_\phi, \Sigma_\phi) \leftarrow q_\phi(z \mid s_t, a_t, r_t)$
5:      Sample latent $z_t \sim \mathcal{N}(\mu_\phi, \Sigma_\phi)$              $\triangleright$ via reparameterization
6:      Decode: $\hat{r}_t \sim p_\theta(r \mid s_t, a_t, z_t)$
7:      ELBO objective:

$$\mathcal{L}_{\text{CVAE}} = \frac{1}{B} \sum_{t=1}^{B} \Big( \log p_\theta(r_t \mid s_t, a_t, z_t) - \beta \, \text{KL}\big( q_\phi(z \mid s_t, a_t, r_t) \, \| \, p(z) \big) \Big)$$

8:      Update $\phi, \theta \leftarrow \phi, \theta + \eta \nabla_{\phi, \theta} \mathcal{L}_{\text{CVAE}}$ with Opt
9: **end while**
10: **return** trained CRM: encoder $q_\phi$, decoder $p_\theta$ (defining $\hat{f}_r(s, a, z)$)

---

**Algorithm 3** Training the Counterfactual State Generator (CSG)

---

**Require:** Trained CRM $(q_\phi, p_\theta)$; generator $g_\psi$; state normalization $\sigma_s$; band radii $(\rho_{\text{low}}, \rho_{\text{high}})$; weights $(\lambda_r, \lambda_{\text{band}})$; optimizer Opt
1: Initialize parameters $\psi$
2: **while** not converged **do**
3:      Sample minibatch $\{(s_t, a_t, r_t)\}_{t=1}^{B}$
4:      Abduction: $z_t \leftarrow \mu_\phi(s_t, a_t, r_t)$          $\triangleright$ posterior mean from CRM encoder
5:      Generate counterfactual proposal: $s_c \leftarrow g_\psi(s_t, a_t, z_t)$
6:      Normalized move: $\Delta_t \leftarrow (s_c - s_t)/\sigma_s$
7:      Reward consistency:
$$L_r = \frac{1}{B} \sum_t \big( \hat{f}_r(s_c, a_t, z_t) - r_t \big)^2$$

8:      Band penalty:

$$L_{\text{band}} = \frac{1}{B} \sum_t \Big( [\rho_{\text{low}} - \|\Delta_t\|_2]_+^2 + [\|\Delta_t\|_2 - \rho_{\text{high}}]_+^2 \Big)$$

9:      Total loss: $\mathcal{L}_{\text{CSG}} = \lambda_r L_r + \lambda_{\text{band}} L_{\text{band}}$
10:     Update $\psi \leftarrow \psi - \eta \nabla_\psi \mathcal{L}_{\text{CSG}}$ with Opt
11: **end while**
12: **return** trained generator $g_\psi$

---

**Algorithm 4** Acceptance gate for a counterfactual proposal

---

**Require:** $(s_t, a_t, r_t)$; posterior mean $z_t = \mu_\phi(s_t, a_t, r_t)$; proposal $s_c = g_\psi(s_t, a_t, z_t)$; thresholds $(\varepsilon_r, \rho_{\text{high}})$; normalization $\sigma_s$
1: Reward proximity: $\Delta r \leftarrow \big| \hat{f}_r(s_c, a_t, z_t) - r_t \big|$
2: Move size: $m \leftarrow \left\| \frac{s_c - s_t}{\sigma_s} \right\|_2$
3: **if** $\Delta r \leq \varepsilon_r$ **and** $m \leq \rho_{\text{high}}$ **then**
4:      **return** ACCEPT
5: **else**
6:      **return** REJECT
7: **end if**

---

---

**Algorithm 5** Offline counterfactual augmentation

---

**Require:** Dataset $\mathcal{D} = \{\tau\}$ with trajectories $\tau = (s_0, a_0, r_0, \dots)$; trained CRM $(q_\phi, p_\theta)$; trained CSG $g_\psi$; per-trajectory budget $K$ (proposals per trajectory); acceptance thresholds $(\varepsilon_r, \rho_{\text{high}})$; normalization $\sigma_s$

1: $\mathcal{D}_{\text{aug}} \leftarrow \mathcal{D}$          $\triangleright$ start from the original dataset
2: **for all** trajectory $\tau \in \mathcal{D}$ **do**
3:      Initialize counter $c \leftarrow 0$
4:      **for all** time indices $t$ in $\tau$ **in random order do**
5:          **if** $c \geq K$ **then break**
6:             Compute $z_t \leftarrow \mu_\phi(s_t, a_t, r_t)$
7:             Propose $s_c \leftarrow g_\psi(s_t, a_t, z_t)$
8:             **if** ACCEPT$((s_t, a_t, r_t), z_t, s_c)$ **then**
9:             Replace $(s_t)$ by $(s_c)$ in a copy of $\tau$ to form $\tilde{\tau}$
10:            Append modified trajectory $\tilde{\tau}$ to $\mathcal{D}_{\text{aug}}$
11:            $c \leftarrow c + 1$
12:          **end if**
13:      **end for**
14: **end for**
15: **return** $\mathcal{D}_{\text{aug}}$

---

**Algorithm 6** Train UCF policy on augmented dataset

---

**Require:** $\mathcal{D}_{\text{aug}}$; hybrid model $\pi_\omega$ with modality-specific conv blocks and a final multi-head attention layer; mask weight $\lambda$; optimizer $\text{Opt}_\omega$

1: **while** not converged **do**
2:      Sample trajectory windows (RTG, states, actions) from $\mathcal{D}_{\text{aug}}$
3:      Encode with modality-specific 1D convolutional blocks (RTG/state/action)
4:      Apply final multi-head self-attention to the token sequence
5:      **Action loss** $\mathcal{L}_{\text{action}}$: negative log-likelihood (discrete) or MSE (continuous) for $a_t$
6:      **Mask loss** $\mathcal{L}_{\text{mask}}$: cross-entropy to target $q_{i,j} \propto \mathbf{1}[j \in S_i]$ where $S_i$ are causal parents (state and RTG at time $t$)
7:      Total loss: $\mathcal{L}_{\text{total}} \leftarrow \mathcal{L}_{\text{action}} + \lambda \mathcal{L}_{\text{mask}}$
8:      Descend $\nabla_\omega \mathcal{L}_{\text{total}}$ with $\text{Opt}_\omega$
9: **end while**
10: **return** $\pi_\omega$

---

# G    IMPLEMENTATION DETAILS

We implement UCF on top of the official Decision Transformer codebase[1], incorporating (i) a *Causal Reward Model* (CRM) trained as a conditional VAE, (ii) a *Counterfactual State Generator* (CSG) trained to edit states under causal constraints, and (iii) a *hybrid causal policy architecture* that combines modality-specific convolutions for local dynamics with a final self-attention layer trained with a causal mask loss. Unless otherwise stated, all experiments follow the D4RL protocol with expert-normalized returns.

## G.1    CAUSAL REWARD MODEL (CRM)

The CRM is parameterized as a conditional variational autoencoder. The encoder $q_\phi(z \mid s, a, r)$ and decoder $p_\theta(r \mid s, a, z)$ are implemented as two-layer MLPs with hidden size 256 and ReLU activations. The latent variable dimension is 32. We optimize the ELBO with $\beta = 0.1$, which balances reward reconstruction accuracy and latent regularization. Larger values (e.g., $\beta = 1.0$) led to worse reward prediction in preliminary runs. We use Adam with a learning rate of $3 \times 10^{-4}$, weight decay $10^{-4}$, and batch size 256. Training runs for $10^6$ steps with early stopping.

---

[1] https://github.com/kzl/decision-transformer

Table 4: Hyperparameters for the Causal Reward Model (CRM).

| Hyperparameter | Value |
|---|---|
| Latent dimension | 16 (MuJoCo) / 32 (Antmaze) |
| Network | 2-layer MLP (ReLU, 256) |
| Optimizer | Adam |
| Learning rate | $3 \times 10^{-4}$ |
| Weight decay | $1 \times 10^{-4}$ |
| Batch size | 256 |
| KL weight $\beta$ | 0.1 |
| Training epochs | 40 |

## G.2 COUNTERFACTUAL STATE GENERATOR (CSG)

The generator $g_\psi(s, a, z)$ is a three-layer MLP (hidden size 256, ReLU). States are normalized by dataset statistics and the move band acts in normalized space. The loss combines reward consistency under fixed $(a, z)$ and a band penalty that encourages $\|\Delta_t\|_2 = \|(s_c - s_t)/\sigma_s\|_2$ to lie in $[\rho_{\text{low}}, \rho_{\text{high}}]$. We train with Adam (learning rate $5 \times 10^{-4}$, batch size 256) for 40 epochs. At acceptance, a counterfactual is kept if $|\hat{r}(s_c, a, z) - r| \leq \varepsilon_r$ (adaptive: $0.1 \times$reward std if unspecified) and $\|\Delta_t\|_2 \leq \rho_{\text{high}}$.

Table 5: Hyperparameters for the Counterfactual State Generator (CSG).

| Hyperparameter | Value |
|---|---|
| Network | 3-layer MLP (ReLU, 256) |
| Optimizer | Adam |
| Learning rate | $5 \times 10^{-4}$ |
| Batch size | 256 |
| Training epochs | 40 |
| Move-band radii | $(\rho_{\text{low}}, \rho_{\text{high}}) = (0.2,\ 0.5)$ |
| Band weight $\lambda_{\text{band}}$ | 0.1 |
| Reward tolerance $\varepsilon_r$ | adaptive ($0.1 \times$std of rewards) |
| Acceptance gate | $|\hat{r}(s_c, a, z) - r| \leq \varepsilon_r,\ \|\Delta_t\|_2 \leq \rho_{\text{high}}$ |
| Training subset | Top-70% reward transitions |

## G.3 HYBRID CAUSAL POLICY ARCHITECTURE

**Tokenization and embeddings** Trajectories are tokenized as repeating triplets $(\hat{G}_t, s_t, a_t)$ with context length $K=15$. We use learned positional embeddings and layer normalization. The model dimension is $d_{\text{model}}=128$ by default and 256 for Hopper-medium and Hopper-medium-replay.

**Local processing** We build three parallel streams (RTG, state, action). Each stream applies depthwise 1D convolutions followed by pointwise projections with residual connections (kernel size 3, stride 1). Outputs are concatenated along the channel axis and projected back to $d_{\text{model}}$.

**Global reasoning** A single multi-head self-attention block operates on the fused sequence. We supervise the final attention block, with all heads constrained to place probability mass only on the direct parents of $a_t$, namely $s_t$ and $\hat{G}_t$, and optionally a short window of recent states. Let $A$ be the attention matrix and $q_{i,\cdot}$ the target distribution over valid parents $S_i$. The mask loss is

$$\mathcal{L}_{\text{mask}} = \frac{1}{L} \sum_{i=1}^{L} \sum_{j \in S_i} -q_{i,j} \, \log(A_{i,j} + \epsilon), \quad \epsilon = 10^{-8}.$$

The total loss is

$$\mathcal{L}_{\text{total}} = \mathcal{L}_{\text{action}} + \lambda_{\text{mask}} \mathcal{L}_{\text{mask}}.$$

Table 6: Hybrid causal architecture: convolutional stream details.

| Component | Setting | Notes |
|---|---|---|
| Streams | RTG / State / Action | Parallel, modality-specific |
| Conv blocks per stream | 2 | Depthwise $3\times1$ + pointwise $1\times1$ |
| Hidden channels | 128 (default) | 256 for Hopper-m / m-r |
| Activation / dropout | GELU / 0.1 | Residual + LayerNorm |
| Fusion | Concat + Linear | To $d_{\text{model}}$ |

Table 7: Hybrid causal architecture: attention and supervision.

| Hyperparameter | Value | Notes |
|---|---|---|
| $d_{\text{model}}$ | 128 (default) / 256 (Hopper-m / m-r) | Embedding size |
| Transformer layers | 1 (final only) | On top of conv fusion |
| Attention heads | 2 | Matches implementation |
| Supervised heads | All heads | Last block only |
| Supervise window $w$ | 0–4 | Default $w=2$ (recent states) |
| Decay for $s_{t-k}$ | $0.6^k$ | Geometric weights |
| Parent tokens | $s_t, \hat{G}_t\ (+\ s_{t-k})$ | Small weight on $a_{t-1}$ optional |
| $\epsilon$ in loss | $10^{-8}$ | Numerical stability |
| Optimizer / LR | Adam / $1\times10^{-4}$ | Linear warmup |

**Policy training protocol.** We use Adam, batch size 64, weight decay $10^{-4}$, dropout 0.1, GELU activations, and a linear warmup schedule. Evaluation follows D4RL. Model selection uses validation return.

**Embedding dimension.** We use an embedding dimension of 256 in `hopper-medium` and `hopper-medium-replay`, and 128 in all other environments. This matches prior findings that higher-dimensional embeddings improve stability in Hopper tasks but provide limited benefit elsewhere.

Table 8: Common hyperparameters of UCF policy (based on DC).

| Hyperparameter | Value |
|---|---|
| Number of layers | 1 |
| Hidden size | 128 (default) / 256 (Hopper-medium / replay) |
| Context length $K$ (MuJoCo / AntMaze) | 20 / 50 |
| Dropout | 0.1 |
| Activation | GELU |
| Learning rate | $10^{-4}$ (MuJoCo, AntMaze) |
| Weight decay | $10^{-4}$ |
| Batch size | 64 |
| Optimizer | Adam |
| Learning rate schedule | Linear warmup |

## G.4    DETAILS FOR SPURIOUS CORRELATION EXPERIMENT

This appendix describes how we construct the spurious datasets and the exact evaluation protocol used in the robustness experiment.

**Dataset construction.** We create spurious variants of `HalfCheetah-medium-v2` and `Hopper-medium-v2` (D4RL) as follows: (1) Load the original offline trajectories. (2) Compute the global median of per-step rewards. (3) Define a binary distractor feature $d_t := \mathbb{I}\{r_t >$

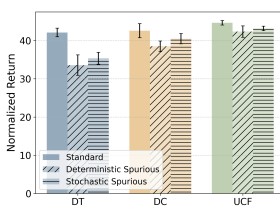 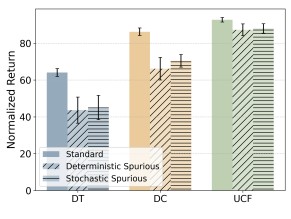 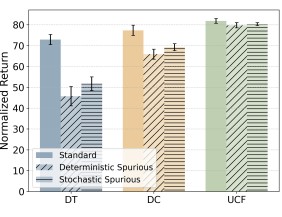

(a) HalfCheetah-medium          (b) Hopper-medium          (c) Walker2d-medium

Figure 4: Robustness to spurious distractors. Expert-normalized returns for DT, DC, and UCF when trained on datasets with deterministic spurious and stochastic spurious.

$\text{median}(r)\}$. (4) Append $d_t$ to each state vector (state dimension increases by one). (5) Save the modified dataset. This makes $d_t$ highly predictive of reward in the training data while having no causal effect on the environment.

**Training protocol.** DT and DC are trained directly on the modified datasets (with $d_t$ appended to states). For UCF, we first apply counterfactual augmentation to the same spurious dataset: the Causal Reward Model (CRM) and Counterfactual State Generator (CSG) generate candidate counterfactual states that are then filtered by the acceptance gates (reward consistency and move-band checks). The distractor $d_t$ is treated as part of the state input but is not used in abduction beyond its presence in $s_t$ and does not enter the reward-preservation objective directly. Accepted counterfactuals are merged with the spurious dataset to form the augmented training set.

**Evaluation protocol.** At test time, we break the learned correlation by a direct intervention in the evaluation loop. Policies are evaluated in the original, unmodified environments. For each observed state, we append a fixed distractor value of $0.0$ before passing it to the policy, matching the training input dimension but removing any predictive content from the distractor.[2]

**Robustness to Stochastic Distractors** In our main robustness experiment (Section 4.4), we utilized a deterministic distractor $d_t = \mathbb{I}[r_t > \text{median}(r)]$, which creates a perfect correlation ($R = 1.0$) between the feature and the reward class. To further validate our method against the reviewer's question regarding noisy sensors, we evaluated all models on a stochastic distractor scenario.

We define a noisy distractor with a flip probability $p = 0.15$:

$$d_t = \mathbb{I}[r_t > \text{median}(r)] \oplus \text{Bernoulli}(0.15) \qquad (6)$$

This distractor preserves the reward-aligned bit with probability 0.85 and randomly flips it with probability 0.15, reducing its reliability while keeping a noticeable correlation with reward. We trained DT, DC, and UCF on this stochastic dataset and evaluated them under the same intervention protocol (fixing $d_t = 0$ at test time).

Figure 4 shows the expert-normalized returns for DT, DC, and UCF under three settings: no distractor, the deterministic distractor, and the stochastic distractor. Adding noise to the spurious feature weakens its predictive value, and the degradation in DT and DC is correspondingly smaller than in the deterministic case. This occurs because the stochastic distractor is a "noisier" predictor than the deterministic one; consequently, the models rely on it slightly less during training, leading to a smaller collapse when the feature is removed. However, their performance remains significantly degraded compared to the standard setting, confirming that they still suffer from causal confusion. In contrast, UCF demonstrates remarkable stability, achieving returns comparable to the clean baseline. These empirical results confirm that UCF's robustness benefits are not limited to deterministic artifacts but extend to stochastic correlations.

---

[2]We use the same feature normalization as in training; the added distractor coordinate is standardized with the training statistics and set to the normalized value corresponding to $d_t{=}0$.

## H  DETAILS ON THE PROPOSED METHOD

**Counterfactual Construction.**  Counterfactual Construction follows the Pearl's three-step causal procedure: abduction, action, and prediction (Pearl et al., 2000; Pearl, 2009).

1. Abduction: Infer the latent disturbance $z_t \sim q_\phi(z \mid s_t, a_t, r_t)$, which captures the unobserved factors that, together with $(s_t, a_t)$, produced the factual reward.

2. Action (Intervention): Define the counterfactual intervention by holding the action $a_t$ fixed. In our reward SCM, $a_t$ is a direct causal parent of $r_t$. To ensure the counterfactual is valid with respect to the reward mechanism, we hold the action parent fixed and solve for a state edit.

3. Prediction: Compute the counterfactual state $s_c = \mathrm{CSG}(s_t, a_t, z_t)$. This step generates the value of the state variable in the counterfactual world, ensuring it remains on-manifold and preserves the reward under the fixed action and abduced context.

In this procedure, $s_t$ provides the factual state as the basis for editing (prediction step), $a_t$ ensures the counterfactual remains consistent with the reward mechanism (intervention step), and $z_t$ ensures the generator respects the unobserved disturbance inferred from evidence (abduction step). Together, these allow UCF to follow the full abduction–action–prediction pipeline and generate valid counterfactual states.

**On the move-band constraint.**  The move band $[\rho_{\mathrm{low}}, \rho_{\mathrm{high}}]$ in Equation (3) serves two purposes beyond reward preservation: (i) *avoiding trivial augmentations* — without a lower bound, the generator can collapse to near-identity edits that add little diversity; and (ii) *preventing off-manifold edits* — without an upper bound, large moves can drift outside dataset support and destabilize policy learning. We normalize state coordinates and apply the constraint in normalized space. Empirically, $(\rho_{\mathrm{low}}, \rho_{\mathrm{high}}) = (0.2, 0.5)$ yields non-trivial yet plausible edits.

**Acceptance gates at generation time.**  After training, we generate counterfactuals and accept $s_c$ only if

$$\left| \hat{f}_r(s_c, a_t, z_t) - r_t \right| \le \varepsilon_r \quad \text{and} \quad \left\| \frac{s_c - s_t}{\sigma_s} \right\|_2 \le \rho_{\mathrm{high}}.$$

The tolerance $\varepsilon_r$ is set adaptively to $0.1 \times \mathrm{std}(r)$ if not specified; only accepted $s_c$ are written back.

**Data augmentation procedure**  For each factual trajectory $\tau = \{(s_t, a_t, r_t)\}_{t=0}^{T-1}$, we create a small number of *augmented copies*. In each copy, we scan time steps that pass a reward quantile filter and propose $s_c = g_\psi(s_t, a_t, \mu_\phi(s_t, a_t, r_t))$. If the candidate passes the two acceptance gates above, we replace only the observation token $s_t$ with the counterfactual $s_c$. The next observation $s_{t+1}$ is kept factual; we do not treat it as the successor of $s_c$. The final dataset on disk is the union of all original trajectories and their edited copies; no in-batch replacement is performed later.

## I  ABLATION STUDIES ON COUNTERFACTUAL GENERATION

We conduct ablations to evaluate the design choices in the counterfactual generation pipeline. Experiments are performed on `hopper-medium-v2` and `halfcheetah-medium-v2`, with results reported as expert-normalized return (mean $\pm$ std over 5 seeds).

### I.1  REWARD CONSISTENCY GATE

The counterfactual generator accepts a proposal only if

$$\left| \hat{f}_r(s_c, a_t, z_t) - r_t \right| \le \varepsilon_r,$$

which ensures counterfactuals remain consistent with the factual reward. Without this gate, counterfactuals may provide contradictory training signals. In implementation, $\varepsilon_r$ is automatically set to $0.1 \times \mathrm{std}(r)$, scaling the tolerance relative to reward variability in the dataset.

Table 9: Ablation on reward consistency gate. Expert-normalized returns (mean ± std over 5 seeds).

| Method | Hopper-medium | HalfCheetah-medium |
|---|---|---|
| UCF (full) | $93.4 \pm 1.5$ | $44.9 \pm 0.7$ |
| w/o reward gate | $74.7 \pm 3.3$ | $42.1 \pm 1.1$ |

We find that removing the gate significantly degrades Hopper performance, with a smaller but consistent drop on HalfCheetah. This confirms that label-preservation under the Causal Reward Model is critical for stable counterfactual training.

## I.2 BAND CONSTRAINT

The move band constrains the normalized edit magnitude $\|\Delta_t\|_2 = \|(s_c - s_t)/\sigma_s\|_2$ to lie within $[\rho_{\text{low}}, \rho_{\text{high}}]$. This discourages both trivial near-copies ($\|\Delta_t\|_2 < \rho_{\text{low}}$) and implausible off-manifold shifts ($\|\Delta_t\|_2 > \rho_{\text{high}}$). We default to $(\rho_{\text{low}}, \rho_{\text{high}}) = (0.2, 0.5)$ and weight the penalty with $\lambda_{\text{band}} = 0.1$.

Table 10 shows that this constraint is critical for performance. Removing the band constraint entirely ($\lambda_{\text{band}} = 0$) causes a significant performance collapse on both environments. Relaxing only the lower bound ($\rho_{\text{low}} = 0$), which allows for trivial copies, also degrades performance, confirming the value of encouraging meaningful state edits.

Table 10: Ablation on the band constraint. Expert-normalized returns (mean ± std over 5 seeds).

| Method | Hopper-medium | HalfCheetah-medium |
|---|---|---|
| UCF (full) | $93.4 \pm 1.5$ | $44.9 \pm 0.7$ |
| w/o band constraint ($\lambda_{\text{band}} = 0$) | $68.2 \pm 4.1$ | $41.9 \pm 1.8$ |
| w/o lower band ($\rho_{\text{low}} = 0$) | $85.9 \pm 2.4$ | $43.7 \pm 0.9$ |

The diagnostics in Figure 5 and Figure 6 explain these choices.

**Move histograms.** The empirical $\|\Delta\|_2$ distributions are unimodal, with modes around 0.33–0.37. This suggests that the generator naturally prefers moderate edits rather than extremely small or large ones. Our default band of $(0.2, 0.5)$ brackets this region, retaining the majority of plausible moves while excluding both trivial near-copies ($< 0.2$) and overly aggressive shifts ($> 0.5$). This balance prevents the generator from collapsing to identity mappings while also discouraging unrealistic counterfactuals.

**Acceptance heatmaps.** We further sweep $(\rho_{\text{low}}, \rho_{\text{high}})$ and report acceptance rates. The heatmaps show that our chosen range $(0.2, 0.5)$ maintains high acceptance (around 90% 95% on Hopper-medium and around 70% 75% for HalfCheetah-medium) while keeping edits within a safe move region. Tightening the lower bound (e.g., $\rho_{\text{low}} = 0.3$) sharply decreases acceptance, discarding many otherwise valid counterfactuals. Conversely, relaxing the upper bound (e.g., $\rho_{\text{high}} > 0.6$) increases acceptance but allows off-manifold edits(because the $\rho_{\text{high}}$ term is used in Acceptance Gate).

**PCA overlays.** To assess the geometric plausibility of generated states, we visualize original and counterfactual states in the top two PCA components. Counterfactuals (red) largely overlap with the support of the original dataset (blue), indicating that edits remain on-manifold under our chosen range $(0.2, 0.5)$. The necessity of the band constraint and the reward consistency gate is paramount to achieving this result; ablation studies confirm that relaxing these constraints leads to off-manifold drift and subsequent degradation in policy performance. This visual confirmation of on-manifold generation validates the safety mechanism of the counterfactual pipeline.

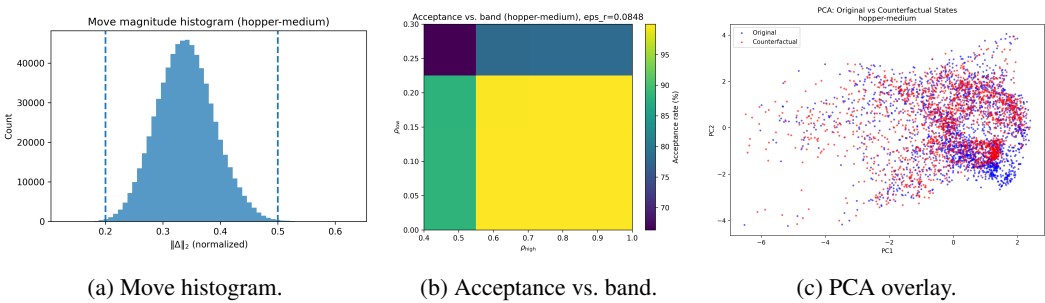

(a) Move histogram.  (b) Acceptance vs. band.  (c) PCA overlay.

Figure 5: **Hopper-medium** diagnostics. The default band $(0.2, 0.5)$ achieves high acceptance while counterfactuals stay on-manifold.

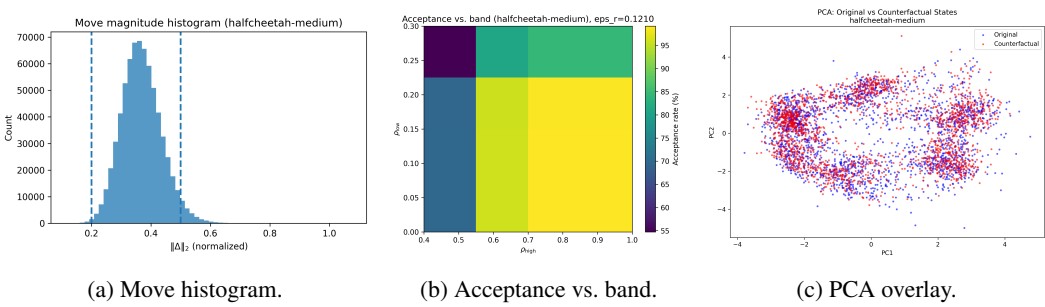

(a) Move histogram.  (b) Acceptance vs. band.  (c) PCA overlay.

Figure 6: **HalfCheetah-medium** diagnostics. The default band $(0.2, 0.5)$ balances coverage and precision; counterfactuals remain aligned with the original manifold.

# J  SHORT-CONTEXT DECISION TRANSFORMER DOES NOT ADDRESS CAUSAL CONFUSION

Several recent works have tested whether reducing the Decision Transformer (DT) context length improves stability. This idea was evaluated directly by the Decision ConvFormer (Kim et al., 2024). In their Appendix G.3 (Table 19), they report that DT performance decreases as the context window is shortened. These findings are consistent with our robustness results in Figure 3 and Figure 4. Our experiments show that the standard DT fails because it overfits to spurious correlations, a problem of causal confusion that is not resolved by simply reducing the context length.

