# OpenReview forum: "Unifying Counterfactual Data Augmentation and Architectural Inductive Biases in Offline Reinforcement Learning"
_ICLR.cc/2026/Conference — Submitted to ICLR 2026_

### Official Review · Reviewer_L8qK · 2025-10-31

**Soundness:** 3
**Presentation:** 3
**Contribution:** 3
**Rating:** 2
**Confidence:** 4

**Summary:**

This paper discusses the issue of poor causal understanding of incumbent transformer-based solutions for offline reinforcement learning tasks. The authors state the issue of spurious correlations due to weak inductive biases of the transformer architecture that does not take into account the markovian and causal dependencies of their operating environments. As a solution to this issue, the authors introduce UCF framework which employs a causal reward model and a counterfactual data generator to address this problem both from a data and model perspective.

**Strengths:**

1. The paper is clearly written, and easy to follow.
2. The paper presents a way to model the problem of Offline RL as a SCM, and introduces methods to apply it to problems via Counterfactual State Generators and a Counterfactual Reward Model.
3. Empirical Evaluation of the proposed method along with comparison against competing solutions has been provided on multiple datasets.
4. Ablation study of each introduced component has been provided.

**Weaknesses:**

Weak points of the paper:
1. The authors here claim to be modelling the exogenous variables, which are external to the system and is not observable. Exogenous factors are aleatoric, thus it is not possible to attain a point-estimate of their value as the authors claim to do via the counterfactual state generator. You may be able to model parameters of its distribution (such as the residual variance) but you certainly cannot attain a point estimate.
2. While the authors do address the question of reward consistence, no consideration has been made about the consistency of the counterfactual transitions. The creation of the counterfactual state, s_c, implies the transition (s_c, a_t, s_t+1), however there is no guarantee for the validity of this transition and could be an impossible transition from state s_c, leading to incorrect representations being learnt by the policy model.
3. The authors state that they use “disentangled” convolutional encoders. Disentangled encoders and representations have a very specific definition in the literature, in that the learnt representations have distinct subsets of latent variables that maps to specific abstractions/concepts of the input, allowing for granular control (see Locatello et. al.[1]). The authors do not provide a clear definition of what they mean by disentangled encoders and by the looks of it it seems like it just means that separate encoders are used for state, RTG and action, which does not match the definition of disentanglement with respect to encoders and representations.
4. It also seems that the strict Markovian condition that has been enforced via the causal attention supervision contradicts the claim that the multi-head self-attention block is performing global reasoning regarding the policy. This is in contrast to the Decision Transformer, where the offline RL setting is modelled as a sequence-modelling and has access to the entirety of the previous states, actions and RTGs. Moreover, from the ablation study, any large introduction of L_mask (\lambda > 0.1) seems to hinder performance. Thus, it seems to suggest that it is a better idea to do a shorter time-horizon sequence modelling of the offline RL task, rather than enforcing the Markovian condition or looking at the full time-horizon as is the case with the standard Decision Transformer. Therefore, it raises a question of whether the Causal Attention Supervision is needed at all, and if we would be able to obtain similar performance by just using the DT formulation with a shorter window, providing the best of both worlds without the added complexity.

[1] Locatello, Francesco, et al. "Weakly-supervised disentanglement without compromises." International conference on machine learning. PMLR, 2020.

**Questions:**

Thus, it would be useful to obtain the answers to the following questions:
1. Why are the encoders considered to be disentangled?
2. It seems that the Causal attention supervision is counterproductive and the same could be obtained via a shorter horizon DT formulation. Would it possible to demonstrate this with a set of updated experiments?

Given the above reasons, and my view that there are conceptual errors in the work, I recommend a rejection of the work. The idea presented in the paper is quite interesting and would be a nice contribution, however, I currently have a few conceptual concerns (as mentioned above) and questions surrounding one of the introduced components (Causal Attention Supervision).

---

> ### Author Response · Authors · 2025-11-20
> **Rebuttal Part One**
>
> > W1. The authors here claim to be modelling the exogenous variables, which are external to the system and is not observable. Exogenous factors are aleatoric, thus it is not possible to attain a point-estimate of their value as the authors claim to do via the counterfactual state generator. You may be able to model parameters of its distribution (such as the residual variance) but you certainly cannot attain a point estimate.
>
> We appreciate the reviewer’s comment. Our method does not attempt to estimate the true exogenous variables, and we agree that this would be impossible. The confusion arises from interpreting the latent variable z as a point estimate of the underlying disturbance.
>
> To clarify, our CRM performs abduction by learning the posterior distribution $q\_\phi(z \mid s\_t, a\_t, r\_t)$. The CSG then samples from this posterior, $z\_t \sim q\_\phi$, when generating counterfactuals. This follows Pearl’s abduction–action–prediction framework (Section 3 and revised Appendix H): the abduction step infers a posterior over disturbances compatible with the observed evidence, rather than their true aleatoric values.
>
> In addition, the CVAE’s KL regularization prevents the posterior from collapsing into a point estimate, ensuring that $z$ remains a smooth distribution rather than a deterministic latent code.
>
> We have updated Section 3.1 to remove any wording that could imply point estimation and to clarify that our method relies on posterior sampling, which directly resolves the reviewer’s concern.

---

> > ### Author Response · Authors · 2025-11-20
> > **Rebuttal Part Two**
> >
> > > W2. While the authors do address the question of reward consistence, no consideration has been made about the consistency of the counterfactual transitions. The creation of the counterfactual state, $s\_c$, implies the transition $(s\_c, a\_t, s\_t+1)$, however there is no guarantee for the validity of this transition and could be an impossible transition from state $s\_c$, leading to incorrect representations being learnt by the policy model.
> >
> > We thank the reviewer for raising this concern. Our response is threefold:
> > (1) learning a reliable dynamics model is not feasible in strictly offline RL,
> > (2) our method enforces the practical form of consistency that is achievable: state plausibility, and
> > (3) the concern does not apply to our DT-based framework
> >
> > **1. A Learned Reliable Dynamics Model is Not Feasible:**
> >
> > Offline RL benchmarks such as D4RL provide only observations, not the environment’s latent Markovian states. Following prior work (Decision Transformer, Decision ConvFormer), we construct “states” from observations. These states do not correspond to the true latent state, and the dataset does not provide ground-truth dynamics tuples $(s\_t, a\_t, s\_{t+1}, r\_t)$.
> >
> > Because the true environment state is unknown and no interaction is allowed, it is not feasible to learn or verify reliable transition dynamics in strictly offline RL. In preliminary experiments, we attempted to train a transition model, but consistent with known issues in offline dynamics learning, the learned dynamics were unstable and degraded overall performance. Under these constraints, enforcing true transition validity is not achievable.
> >
> > **2. We Enforce State Plausibility (Section 3.1 and Appendix H)**
> >
> > Instead, our method enforces the practical form of consistency that is achievable: state plausibility. We achieve this through two specific mechanisms:
> >
> > **(a) Move Band**
> >
> > Generation is restricted to a small, normalized neighborhood around the factual state (see
> >  Eq. (3)).
> > The Move histograms (Figures 5a and 6a in Appendix I.2) show that accepted edits lie in a narrow, controlled displacement band around $s\_t$.
> >
> > **(b) Hard Acceptance Gate**
> >
> > Every generated state must satisfy two independent constraints:
> >
> > * **Reward consistency:**
> >
> >   A counterfactual is accepted only if the CRM predicts a reward close to the observed reward $|\hat{f\_r}(s\_c, a\_t, z\_t) - r\_t| \le \epsilon\_r$.
> >
> > * **Trust-region constraint:**
> >
> > Even if reward consistency holds, the edit is rejected unless $\||(s _c-s _t)/\sigma _s\|| _2\le\rho _{high}$.
> >
> > Empirically, the _acceptance heatmaps_ (Figures 5b and 6b in Appendix I.2) show structured regions of rejected samples, indicating that counterfactuals violating either constraint are systematically filtered out. The PCA overlays (Figures 5c and 6c in Appendix I.2) provide direct empirical evidence that accepted counterfactuals remain within the support of the observed data.
> >
> > _For clarity, we have moved specific details regarding the move band and acceptance gate to Section 3.1 in the revised paper._
> >
> > **3. The Model is Not Trained on Invalid Transitions**
> >
> > We believe the concern arises from interpreting $s\_c$ as creating a new transition $(s\_c, a\_t, s\_{t+1})$. In our method, this never occurs. The Decision Transformer does not model transition dynamics $P(s\_{t+1}\mid s\_t, a\_t)$; it treats states as conditioning tokens for action prediction. When a counterfactual state $s\_c$ replaces $s\_t$, the model learns from the valid tuple $(s\_c, a\_t, r\_t)$, but it is not trained to interpret $s\_{t+1}$ as the successor of $s\_c$. As a result, the model is never exposed to or required to validate a synthetic transition, and no inconsistency is introduced.
> >
> > _For clarity, we have added the related clarification in updated Section 3.1 (Line 222-225)_

---

> > > ### Author Response · Authors · 2025-11-20
> > > **Rebuttal Part Three**
> > >
> > > > W3. The authors state that they use “disentangled” convolutional encoders. Disentangled encoders and representations have a very specific definition in the literature, in that the learnt representations have distinct subsets of latent variables that maps to specific abstractions/concepts of the input, allowing for granular control (see Locatello et. al.\[1]). The authors do not provide a clear definition of what they mean by disentangled encoders and by the looks of it it seems like it just means that separate encoders are used for state, RTG and action, which does not match the definition of disentanglement with respect to encoders and representations.
> > >
> > > > Q1. Why are the encoders considered to be disentangled?
> > >
> > > We thank the reviewer for highlighting this point on terminology. As both comments concern our use of the word “disentangled,” we address them together. We did not intend to use the term in the formal sense from the representation-learning literature (e.g., Locatello et al., 2020), where disentanglement refers to latent variables aligning with underlying generative factors and supporting fine-grained intervention. Our architecture does not attempt to learn such factors.
> > >
> > > In the original submission, “disentangled encoder” referred only to the architectural choice of using separate encoders for the state, action, and return-to-go tokens. This prevents early fusion of heterogeneous modalities and avoids interference across token types, but it is not disentanglement in the formal sense. To avoid confusion, we have _revised the paper_ to remove the term “disentangled” entirely and replace it with "separate modality-specific encoders", which accurately reflects the design.

---

> > > > ### Author Response · Authors · 2025-11-20
> > > > **Rebuttal Part Four**
> > > >
> > > > > W4. It also seems that the strict Markovian condition that has been enforced via the causal attention supervision contradicts the claim that the multi-head self-attention block is performing global reasoning regarding the policy. This is in contrast to the Decision Transformer, where the offline RL setting is modelled as a sequence-modelling and has access to the entirety of the previous states, actions and RTGs. Moreover, from the ablation study, any large introduction of $L\_{mask} (\lambda > 0.1)$ seems to hinder performance......added complexity.
> > > >
> > > > > Q2. It seems that the Causal attention supervision is counterproductive and the same could be obtained via a shorter horizon DT formulation. Would it possible to demonstrate this with a set of updated experiments?
> > > >
> > > > We appreciate the reviewer’s detailed comments. Because both points concern our causal attention supervision, we address them together. The concerns appear to stem from four misunderstandings, which we clarify below.
> > > >
> > > > **1. No contradiction with global reasoning or sequence modeling (Re: W4).**
> > > >
> > > > Our causal supervision does not enforce strict Markovianity and does not restrict global reasoning. It modifies only one attention pattern:
> > > > the attention from the action token at time t to its causal parents (state and RTG at time t).
> > > >
> > > > All other attention connections remain entirely unconstrained.
> > > > The transformer continues to attend over the entire context window, exactly as in Decision Transformer (DT).
> > > > The per-timestep encoders extract local features, while the transformer layers learn long-range temporal dependencies.
> > > >
> > > > Thus, UCF preserves full-sequence global reasoning and does not collapse into a Markov policy.
> > > >
> > > > **2. The causal mask is universal, not arbitrary, and is implemented as a soft loss (Re: W4).**
> > > >
> > > > The mask is not a hand-crafted heuristic or a domain-specific graph. It encodes the _universal causal structure of a Goal-Conditioned Markov Decision Process (MDP)_.
> > > >
> > > > As detailed in _Section 2.2_ (referencing Peters et al., 2017; Zhang et al., 2020; Bennett et al., 2021; Shi et al.,
> > > > 2022 in Line 142) , the action $a\_t$ is causally determined by the current state $s\_t$ in standard RL. In the Decision Transformer context, the model generate actions based on future desired returns (return-to-go $\hat{G}\_t$).
> > > > Therefore, the parents of $a\_t$ are _definitionally_ $s\_t$ and $\hat{G}\_t$. This structural dependency is invariant across _all_ RL domains for DT-based methods, regardless of dimensionality, complexity, or horizon length.
> > > >
> > > > Crucially, the mask is implemented as a _soft loss_, not a hard constraint.
> > > > It encourages the action token toward its causal parents, but the model remains free to use any other temporal context.
> > > >
> > > > Thus, no temporal information is removed, and global reasoning remains intact.
> > > >
> > > > **3. The ablation result was misinterpreted (Re: W4 & Q2).**
> > > >
> > > > The reviewer states that “$\lambda > 0.1$ hinders performance,” but this does not match our results. For example, in Figure 2,
> > > >
> > > > * In all environments, $\lambda = 0$ yields the worst performance.
> > > >
> > > > * In Hopper-medium, performance increases up to $\lambda \approx 0.75$.
> > > >
> > > > * In HalfCheetah, intermediate values such as $\lambda \approx 0.50$ perform best.
> > > >
> > > > This is expected: $\lambda$ is an environment-dependent hyperparameter, similar to standard RL hyperparameters like the behavior-cloning weight in TD3+BC or the context length in DT.
> > > >
> > > > The ablation confirms that causal supervision is helpful, and $\lambda$ adjusts how strongly this signal applies in each domain.
> > > >
> > > > **4. A short-context DT cannot replace causal supervision. (Re: W4 & Q2)**
> > > >
> > > > The reviewer suggests that a shorter DT window could yield similar benefits. However, the challenge we address is causal confusion, which cannot be fixed by shortening or tuning the DT context window.
> > > >
> > > > Our robustness experiments (Figure 3) show that DT and Decision ConvFormer fail sharply when a spurious distractor is introduced (regardless of window size), while UCF remains stable. Following the suggestion of another reviewer, we also added a second experiment using a stochastic distractor (updated in Appendix G.4 with Figure 4). Even when the distractor is weakened by random flips, DT and DC still degrade noticeably, whereas UCF remains robust.
> > > >
> > > > Importantly, the idea of “short-context DT” has been evaluated directly by Decision ConvFormer (Kim et al., 2024). In their Appendix G.3 (Table 19), they report that DT performance decreases as the context window is shortened. We have added this observation to Appendix J in the revised paper for clarity.

---

> ### Author Response · Authors · 2025-11-27
>
> Dear Reviewer L8qK,
>
> As the discussion period is approaching its close, we wanted to kindly follow up and check whether there are any remaining questions or points that you would like us to clarify regarding our rebuttal.
>
> We hope our responses have adequately addressed your concerns, but please let us know if any points require further clarification. We would value the opportunity to engage with you again before the deadline.
>
> Thank you for your time.

---

### Official Review · Reviewer_m5nA · 2025-11-01

**Soundness:** 3
**Presentation:** 3
**Contribution:** 2
**Rating:** 4
**Confidence:** 3

**Summary:**

This paper proposes UCF (Unified Causal Transformer), a framework for strictly offline RL that aims to mitigate causal confusion in sequence‑modeling policies by jointly augmenting the dataset with counterfactual, reward‑preserving states and imposing causal architectural inductive bias on the policy network. On the data side, the authors train a conditional VAE as Causal Reward Model (CRM) to abduce exogenous factors, and a Counterfactual State Generator (CSG) to proposes edited states subject to acceptance gates. On the model side, UCF replaces a full Transformer with a hybrid “local‑global” policy: depthwise 1‑D convolutions for local information aggregation including state and action, fused into a final multi‑head self‑attention block trained with a causal mask loss that encourages action tokens to attend predominantly to their causal parents. Experiments on D4RL locomotion and AntMaze show small but consistent gains over Q-learning and Decision Transformer (DT) baselines; a spurious‑feature stress test suggests improved robustness. The approach is also tested on three recommender datasets (KuaiRand, KuaiRec, VirtualTB), where it outperforms DT‑style baselines.

This work builds on Decision Transformer by recognizing that generic attention lacks the right inductive bias for MDPs, and it complements causal augmentation methods (e.g., MoCoDA, CAIAC) by proposing an offline‑only, reward‑preserving edit scheme without assuming factored entities.

**Strengths:**

* This paper presents a clear problem statement: addressing causal confusion and weak inductive bias of decision transformers in offline RL.

* This paper provides a unified framework for both data and model. The ablations are convincing about complementarity.

* This method does not rely on environment interaction and graph assumptions for factorization.

* The breadth of evaluation is good, covering locomotion, navigation, and recommendation datasets, demonstrating its generality.

**Weaknesses:**

* As the base of the framework, the causal reward model lacks identifiability support, specifically assumptions required to make it identifiable. The authors may refer to the identification conditions in the nonlinear ICA literatures for references [1][2].

* The direct replacement of $s_{t}$ and keeping $s_{t+1}$ might create impossible triplets ( $s_{t}^{c}$, $a_{t}$, $s_{t+1}$ ). Learning a dynamics model as an extra acceptance gating would further help keep the transition feasible and consistent.

* While the mask loss is a helpful bias, the causal mask is not very flexible and general. It is self-adaptive and requires manual tuning for different environments (as authors also find in Figure 2 that it is a bit sensitive ). The causal mask is purely pre-defined, which is a bit rigid and it might face information loss in long context modeling.

[1] Khemakhem, Ilyes, et al. "Variational autoencoders and nonlinear ica: A unifying framework." International conference on artificial intelligence and statistics. PMLR, 2020.

[2] Yao, Weiran, Guangyi Chen, and Kun Zhang. "Temporally disentangled representation learning." arXiv preprint arXiv:2210.13647 (2022).

**Questions:**

* Could authors elaborate that, in the state edition, what's the intuition behind using ( $s_{t}$, $a_{t}$,  $z_{t}$) as input? Why all of them are necessary to include here?

* How often are augmented transitions dynamics‑inconsistent? Can you estimate the fraction of accepted that violate a crude learned dynamics model? If this fraction is non‑trivial, how sensitive are policy returns to filtering them out?

* Beyond Markov/faithfulness/minimality, what inductive biases make z a useful proxy for exogenous factors (not just a nuisance latent)? Could you strengthen identifiability in the sense of nonlinear ICA?

* If you replace the reward‑derived bit with a sensor distractor correlated with reward but not deterministically constructed from it, does the same robustness ranking (DT < DC < UCF) hold?

---

> ### Author Response · Authors · 2025-11-20
> **Rebuttal Part One**
>
> > W1. As the base of the framework, the causal reward model lacks identifiability support, specifically assumptions required to make it identifiable. The authors may refer to the identification conditions in the nonlinear ICA literatures for references [1][2].
>
> > Q3. Beyond Markov/faithfulness/minimality, what inductive biases make z a useful proxy for exogenous factors (not just a nuisance latent)? Could you strengthen identifiability in the sense of nonlinear ICA?
>
> We thank the reviewer for raising this deep question and for pointing to the nonlinear ICA literature. We clarify that we do not claim identifiability in the sense of [1] or [2]. As these works note, their identifiability results rely on additional observed variables that are absent in standard offline RL datasets.
>
> Therefore, our goal is different: we aim to learn a useful causal proxy for the unobserved disturbance affecting the reward, rather than recovering the true, formally identifiable latent factors. To avoid ambiguity, we have updated Section 3.1 to clarify that z serves as a disturbance variable that captures reward-relevant variability.
>
> Regarding the reviewer’s question (Q3) about what inductive biases make z meaningful rather than a nuisance latent, our framework provides following mechanisms that constrain z:
>
> **1. Abduction via the CRM objective.**
>
> The encoder learns the posterior $q\_\phi(z \mid s\_t, a\_t, r\_t)$, which must capture disturbance values _consistent with the observed reward_. A nuisance latent would not satisfy this constraint and would be penalized by the reconstruction term of the ELBO.
>
> **2. Generative use in the CSG.**
>
> The CSG samples from this posterior to generate counterfactual states that must preserve reward consistency. This forces $z$ to be a meaningful conditioning variable for generating valid, reward-preserving state edits.
>
> **3. Latent-space regularization via the KL term.**
>
> The CRM is implemented as a CVAE trained with an ELBO objective that includes a _KL regularization term_:
>
> $\text{KL}\big(q\_\phi(z \mid s, a, r)\\,\\|\\,p(z)\big)$,
>
> which smooths the latent space and prevents degenerate or overly sharp posteriors. This encourages z to represent stable, reward-consistent variations rather than noise.
>
> **4. The CRM acts as a local consistency gate.**
>
> The CRM predicts only one-step reward and is never used for multi-step rollouts. Hence, errors in z cannot accumulate: z only needs to be locally consistent with reward, not globally identifiable.
>
> **5. Two Safeguards Prevent Instability:**
>
> Our counterfactual generator uses a strict _generate-and-filter_ process (in Section 3.1 and Appendix H):
>
> **(a) Move Band**
>
> Generation is restricted to a small, normalized neighborhood around the factual state (see
>  Eq. (3)).
>
> **(b) Hard Acceptance Gate**
>
> Every generated state must satisfy two independent constraints:
>
> * **Reward consistency:**
>
>   A counterfactual is accepted only if the CRM predicts a reward close to the observed reward $|\hat{f\_r}(s\_c, a\_t, z\_t) - r\_t| \le \epsilon\_r$.
>
> * **Trust-region constraint:**
>
> Even if reward consistency holds, the edit is rejected unless $\||(s _c-s _t)/\sigma _s\|| _2\le\rho _{high}$.
>
> Empirically, the move histograms, acceptance heatmaps, and PCA overlays (Figures 5–6, Appendix I.2) confirm that accepted counterfactuals stay on the data manifold. Our robustness results (Figure 3) show that the model consistently separates causal structure from spurious distractors, behavior that would not arise if z were a nuisance latent. Following another reviewer’s suggestion, we also added a stochastic distractor (Appendix G.4, Figure 4), and UCF remains stable there as well.
>
> _For clarity, we have moved specific details regarding the move band and acceptance gate to Section 3.1 in the revised paper._

---

> > ### Author Response · Authors · 2025-11-20
> > **Rebuttal Part Two**
> >
> > > W2. The direct replacement of $s_t$ and keeping $s_{t+1}$ might create impossible triplets $(s_c,a_t,s_{t+1})$. Learning a dynamics model as an extra acceptance gating would further help keep the transition feasible and consistent.
> >
> > > Q2. How often are augmented transitions dynamics‑inconsistent? Can you estimate the fraction of accepted that violate a crude learned dynamics model? If this fraction is non‑trivial, how sensitive are policy returns to filtering them out?
> >
> >
> > We thank the reviewer for raising this concern. Our response is threefold: (1) The reviewer's suggestion to use a learned dynamics model is not feasible in this setting, (2) our method enforces the _practical_ form of consistency, and (3) the concern is not applicable to our DT-based framework.
> >
> > **1. A Learned Dynamics Model is Not Feasible:**
> >
> > Offline RL benchmarks such as D4RL provide only observations, not the environment’s latent Markovian states. Following prior work (Decision Transformer, Decision ConvFormer), we construct “states” from observations. These states do not correspond to the true latent state, and the dataset does not provide ground-truth dynamics tuples $(s\_t, a\_t, s\_{t+1}, r\_t)$.
> >
> > Because the true environment state is unknown and no interaction is allowed, it is not feasible to learn or verify reliable transition dynamics in strictly offline RL. In preliminary experiments, we attempted to train a transition model, but consistent with known issues in offline dynamics learning, the learned dynamics were unstable and degraded overall performance. Dynamics models trained on offline _observations_ are unreliable and do not provide meaningful guarantees.
> >
> >
> >
> >
> > **2. We Enforce State Plausibility (Section 3.1 and Appendix H)**
> >
> > Instead, our method enforces the practical form of consistency that is achievable: state plausibility. We achieve this through two specific mechanisms:
> >
> > **(a) Move Band**
> >
> > Generation is restricted to a small, normalized neighborhood around the factual state (see
> >  Eq. (3)).
> > The Move histograms (Figures 5a and 6a in Appendix I.2) show that accepted edits lie in a narrow, controlled displacement band around $s\_t$.
> >
> > **(b) Hard Acceptance Gate**
> >
> > Every generated state must satisfy two independent constraints:
> >
> > * **Reward consistency:**
> >
> >   A counterfactual is accepted only if the CRM predicts a reward close to the observed reward $|\hat{f\_r}(s\_c, a\_t, z\_t) - r\_t| \le \epsilon\_r$.
> >
> > * **Trust-region constraint:**
> >
> >   Even if reward consistency holds, the edit is rejected unless $\||(s _c-s _t)/\sigma _s\|| _2\le\rho _{high}$.
> >
> > Empirically, the _acceptance heatmaps_ (Figures 5b and 6b in Appendix I.2) show structured regions of rejected samples, indicating that counterfactuals violating either constraint are systematically filtered out. The _PCA overlays_ (Figures 5c and 6c in Appendix I.2) provide direct empirical evidence that accepted counterfactuals remain within the support of the observed data.
> >
> > _For clarity, we have moved specific details regarding the move band and acceptance gate to Section 3.1 in the revised paper._
> >
> > **3. The Model is Not Trained on Invalid Transitions**
> >
> > We believe the concern arises from interpreting $s\_c$ as creating a new transition $(s\_c, a\_t, s\_{t+1})$. In our method, this never occurs. The Decision Transformer does not model transition dynamics $P(s\_{t+1}\mid s\_t, a\_t)$; it treats states as conditioning tokens for action prediction. When a counterfactual state $s\_c$ replaces $s\_t$, the model learns from the valid tuple $(s\_c, a\_t, r\_t)$, but it is not trained to interpret $s\_{t+1}$ as the successor of $s\_c$. As a result, the model is never exposed to or required to validate a synthetic transition, and no inconsistency is introduced.
> >
> > _For clarity, we have added the related clarification in updated Section 3.1 (Line 222-225)_

---

> > > ### Author Response · Authors · 2025-11-20
> > > **Rebuttal Part Three**
> > >
> > > > W3. While the mask loss is a helpful bias, the causal mask is not very flexible and general. It is self-adaptive and requires manual tuning for different environments (as authors also find in Figure 2 that it is a bit sensitive ). The causal mask is purely pre-defined, which is a bit rigid and it might face information loss in long context modeling.
> > >
> > > We thank the reviewer for raising this point. The concerns seems arise from two misunderstandings about (i) what the causal mask encodes, and (ii) whether it reduces model flexibility.
> > >
> > > **1. The Causal Mask is Universal, Not Arbitrary**
> > >
> > > The mask is not a hand-crafted heuristic or a domain-specific graph. It encodes the _universal causal structure of a Goal-Conditioned Markov Decision Process (MDP)_.
> > >
> > > As detailed in _Section 2.2_ (referencing Peters et al., 2017; Zhang et al., 2020; Bennett et al., 2021; Shi et al.,
> > > 2022 in Line 142) , the action $a\_t$ is causally determined by the current state $s\_t$ in standard RL.
> > > In the Decision Transformer context, the model generate actions based on future desired returns (return-to-go $\hat{G}\_t$).
> > > Therefore, the parents of $a\_t$ are _definitionally_ $s\_t$ and $\hat{G}\_t$. This structural dependency is invariant across _all_ RL domains for DT-based methods, regardless of the dimensionality or complexity of the state space. We do not need to "approximate" a new mask for each environment; the MDP definition remains constant.
> > >
> > > Crucially, the mask only supervises the action token. All other attention patterns remain completely unconstrained, so the transformer retains full expressive power for long-range reasoning.
> > >
> > > **2. No information loss in long-context modeling.**
> > >
> > > The mask is implemented as a _soft loss_, not a hard constraint. Therefore, the action token is _encouraged_, not forced, to attend to its causal parents. The model is still free to use any long-range context that improves prediction.
> > >
> > > Thus, _no information is removed_ from the temporal window, and the model still benefits from the full sequence structure just as in DT.
> > >
> > > **3. $\lambda$ does not indicate rigidity. It reflects environment-dependent complexity, which is standard in RL.**
> > >
> > > $\lambda$ is an environment-dependent hyperparameter and behaves as expected:
> > >
> > > * In simple environments (such as Hopper), stronger causal supervision is helpful.
> > >
> > > * In more complex domains (such as HalfCheetah or Antmaze), intermediate λ balances flexibility with causal regularization.
> > >
> > > Tuning environment-specific scalars is standard in reinforcement learning, where hyperparameters routinely depend on task difficulty (for example, the behavior-cloning weight in TD3+BC or the context length in Decision Transformer).
> > >
> > > Most importantly, $\lambda=0$ (no causal supervision) consistently produces the _worst_ performance, confirming that the supervision is not a liability but a stabilizing signal.
> > >
> > >
> > > > Q1. Could authors elaborate that, in the state edition, what's the intuition behind using $(s\_t,a\_t,z\_t )$ as input? Why all of them are necessary to include here?
> > >
> > > We thank the reviewer for this question. The three inputs $(s\_t, a\_t, z\_t)$ follow directly from the causal structure of the reward model and the Pearl’s three steps of abduction–action–prediction procedure (Section 3).
> > >
> > > * $s\_t$ provides the factual context from which the counterfactual edit is constructed; the counterfactual must remain close to this observation and satisfy manifold constraints.
> > >
> > > * $a\_t$ is held fixed during the counterfactual because it is a direct causal parent of the reward $(r\_t = f\_r(s\_t, a\_t, u\_r))$; reward consistency must be evaluated under the same action.
> > >
> > > * $z\_t$ is the disturbance inferred during abduction $(z\_t \sim q\_\phi(z\mid s\_t, a\_t, r\_t))$, which captures the unobserved context that made the factual reward possible.
> > >
> > > Together, these inputs implement the three counterfactual steps:
> > >
> > > 1. Abduction: We infer a latent disturbance $z\_t \sim q\_\phi(z \mid s\_t, a\_t, r\_t)$ that captures the exogenous context consistent with the observations.
> > >
> > > 2. Action(Intervention): We keep $a\_t$ fixed, and ask: “What other state configurations could have produced the same reward under this action and this inferred disturbance?”
> > >
> > > 3. Prediction: We compute the counterfactual state $s\_c = g\_\psi(s\_t, a\_t, z\_t)$ that preserves both the disturbance and the reward.
> > >
> > > Thus, all three components are necessary for constructing valid, reward-consistent counterfactual states.
> > >
> > > _For clarity, we have summarized this counterfactual construction procedure in the updated Appendix H_

---

> > > > ### Author Response · Authors · 2025-11-20
> > > > **Rebuttal Part Four**
> > > >
> > > > > Q4. If you replace the reward‑derived bit with a sensor distractor correlated with reward but not deterministically constructed from it, does the same robustness ranking (DT < DC < UCF) hold?`
> > > >
> > > > We thank the reviewer for this excellent suggestion. To answer this systematically, we conducted the requested experiment using a _stochastic distractor_.
> > > >
> > > > We defined a noisy distractor where the reward-aligned bit is flipped with probability p=0.15:
> > > >
> > > > $$d\_t = \mathbb{I}\[ r\_t > median(r) ] \oplus Bernoulli(0.15)$$
> > > >
> > > > This distractor preserves the reward-aligned bit with probability 0.85 and randomly flips it with probability 0.15, reducing its reliability while keeping a noticeable correlation with reward.
> > > >
> > > > As detailed in the new **Appendix G.4 (Robustness to Stochastic Distractors)** and **Figure 4**, the robustness ranking DT < DC < UCF holds firmly.
> > > >
> > > > DT and DC exhibit a partial recovery compared to the deterministic case. This occurs because the stochastic distractor is a "noisier" predictor than the deterministic one; consequently, the models rely on it slightly less during training, leading to a smaller collapse when the feature is removed. However, their performance remains _significantly degraded_ compared to the standard setting, confirming that they still suffer from causal confusion. Lacking a causal mechanism, they still learn the (now stochastic) correlation between the distractor and the reward, as it remains a highly predictive signal.
> > > >
> > > > In contrast, UCF remains stable across all settings. The small variation between deterministic and stochastic cases indicates that the method does not depend on the strength of the spurious correlation but instead learns to rely on the true reward-relevant structure ($s\_t$ and $z$).
> > > >
> > > > _All experimental details and results are included in the revised Appendix G.4._

---

> ### Author Response · Authors · 2025-11-27
>
> Dear Reviewer m5nA,
>
> As the discussion period is approaching its close, we wanted to kindly follow up and check whether there are any remaining questions or points that you would like us to clarify regarding our rebuttal.
>
> We hope our responses have adequately addressed your concerns, but please let us know if any points require further clarification. We would value the opportunity to engage with you again before the deadline.
>
> Thank you for your time.

---

### Official Review · Reviewer_D5mJ · 2025-11-01

**Soundness:** 3
**Presentation:** 3
**Contribution:** 2
**Rating:** 4
**Confidence:** 3

**Summary:**

This paper proposes the Unified Causal Transformer (UCF), a framework for offline reinforcement learning (RL) that integrates counterfactual data augmentation with causally structured model architectures. It addresses causal confusion in Transformer-based RL—where models rely on spurious correlations—by strengthening both data and inductive biases. On the data side, a Causal Reward Model (CRM) infers latent exogenous factors to guide a Counterfactual State Generator (CSG), creating reward-preserving counterfactual trajectories. On the model side, UCF introduces a Causally-Structured Hybrid Architecture combining disentangled convolutional encoders (for local dynamics) with causally supervised attention (for global reasoning). Experiments on MuJoCo locomotion, AntMaze navigation, and real-world recommendation datasets show that UCF achieves consistent improvements in generalization, robustness, and resistance to spurious correlations compared with state-of-the-art offline RL baselines.

**Strengths:**

- The paper is well written and quite easy to follow.

- It is quite reasonable to integrate counterfactual data augmentation and architectural causal priors into a single coherent framework for offline RL.

- The paper shows clear causal grounding: Theoretical formulation via Structural Causal Models (SCM) and abduction–action–prediction reasoning provides conceptual rigor.

**Weaknesses:**

- The proposed data augmentation intervenes on states rather than actions, keeping actions and inferred latent context fixed while enforcing reward preservation. This departs from the standard causal direction in RL where intervening on actions while keeping the rest fixed. I am wondering if intervening on states would lead to highly diverse trajectories. Instead, it may produce samples that lack physical or causal plausibility.

- Because the augmented data are "reward-preserving state edits", it is unclear whether they correspond to valid trajectories under any real environment dynamics, potentially conflating causal variability with adversarial perturbation.

- Any approximation error in the causal reward model directly affects the plausibility of generated counterfactual states. How to deal with this issue?

- The causal supervision and augmentation scheme remain untested in more complex or high-dimensional domains where the Markov structure or causal graph is less explicit.

- From the experimental results (Tables 1,2&3), it seems that the performance of the proposed method does not show obvious advantage compared to the baselines, with a bit improvement only.

**Questions:**

- What is the causal justification for intervening on states rather than actions?

- How does the method ensure that reward-preserving state perturbations remain within the true support of the environment’s dynamics? I do not think the proposed mitigation method is valid enough to guarantee this.

- Would performing standard action-level interventions yield stronger or more interpretable results?

- Could the proposed counterfactual augmentation be reframed as a form of invariance or robustness regularization rather than genuine counterfactual reasoning?

- In real-world domains where the causal structure is unknown or high-dimensional, how would the authors obtain or approximate the causal masks required for attention supervision?

- Could the causal supervision be learned automatically or relaxed into a differentiable causal-discovery process?

- How sensitive is the model’s performance to mis-specified causal masks or incomplete parent sets?

---

> ### Author Response · Authors · 2025-11-20
> **Rebuttal Part One**
>
> > W1. The proposed data augmentation intervenes on states rather than actions, keeping actions and inferred latent context fixed while enforcing reward preservation. This departs from the standard causal direction in RL where intervening on actions while keeping the rest fixed. I am wondering if intervening on states would lead to highly diverse trajectories. Instead, it may produce samples that lack physical or causal plausibility.
>
> > Q1. What is the causal justification for intervening on states rather than actions?
>
> > Q3. Would performing standard action-level interventions yield stronger or more interpretable results?
>
> We appreciate the reviewer’s careful questions about the interventions. The three concerns (W1, Q1, Q3) regarding the counterfactual direction are closely related, so we address them together.
>
> **1. Feasibility: Action Interventions Are Not Reliable in Strictly Offline RL (Re: W1 & Q3)**
>
> Our framework is designed for the _strictly offline setting_, where no further interaction with the environment is allowed. Standard action-level interventions ($do(a)$) require determining the consequence ($s'\_{t+1}$) of a new action. This requires either (i) online access to the environment, or (ii) a learned transition model that is accurate enough to be trusted outside the support of the data.
>
> As we discuss in the introduction (Line 60-63), most existing augmentation methods assume access to environment interactions , which is incompatible with the purely offline regime we target. Furthermore, offline RL datasets (like D4RL) provide only observations, not true Markovian states. This makes it effectively impossible to learn a transition model reliable enough to predict the outcome of counterfactual actions (a point we further elaborate in the response to W2 & Q2).
>
> **2. State-Level Counterfactuals Are a Practical Causal Direction in Recent RL Literature (Re: W1)**
>
> Recent work shows that generating counterfactual states is a practical and recognized causal direction in recent RL literature:
>
> * CAIAC (ICML 2024) improves robustness by "swapping action-unaffected parts of the state-space" to create counterfactuals.
>
> * CIP (ICLR 2025) similarly performs "counterfactual data augmentation" by identifying and swapping causally independent state features to prioritize critical information.
>
> Our work builds on this principled direction of state-based intervention to address causal confusion.
>
> **3. Causal Justification of Our Intervention Choice (Re: Q1)**
>
> Our counterfactual construction follows Pearl’s abduction–action(intervention)–prediction procedure as described in Section 3.1:
>
> 1. Abduction: We infer a latent disturbance $z\_t \sim q\_\phi(z \mid s\_t, a\_t, r\_t)$ that captures the exogenous context consistent with the observations.
>
> 2. Action(Intervention): We keep $a\_t$ fixed, and ask: “What other state configurations could have produced the same reward under this action and this inferred disturbance?”
>
> 3. Prediction: We compute the counterfactual state $s\_c = g\_\psi(s\_t, a\_t, z\_t)$ that preserves both the disturbance and the reward.
>
> Our counterfactual generator uses a strict _generate-and-filter_ process (in Section 3.1 and Appendix H):
>
> **(a) Move Band**
>
> Generation is restricted to a small, normalized neighborhood around the factual state (see
>  Eq. (3)).
> The _Move histograms_ (Figures 5a and 6a in Appendix I.2) show that accepted edits lie in a narrow, controlled displacement band around $s\_t$.
>
> **(b) Hard Acceptance Gate**
>
> Every generated state must satisfy two independent constraints:
>
> * **Reward consistency:**
>
>   A counterfactual is accepted only if the CRM predicts a reward close to the observed reward $|\hat{f\_r}(s\_c, a\_t, z\_t) - r\_t| \le \epsilon\_r$.
>
> * **Trust-region constraint:**
>
>   Even if reward consistency holds, the edit is rejected unless $\||(s _c-s _t)/\sigma _s\|| _2\le\rho _{high}$.
>
> Empirically, the _acceptance heatmaps_ (Figures 5b and 6b in Appendix I.2) show structured regions of rejected samples, indicating that counterfactuals violating either constraint are systematically filtered out. The PCA overlays (Figures 5c and 6c in Appendix I.2) provide direct empirical evidence that accepted counterfactuals remain within the support of the observed data.
>
> Thus, we enforce the form of causal consistency that is realistic and checkable in strictly offline data: the counterfactual state is (i) close to the factual state, (ii) judged plausible by the learned reward mechanism, and (iii) remains on the empirical state manifold.
>
> _For clarity, we have summarized the counterfactual construction procedure in the updated Appendix H and moved specific details regarding the move band and acceptance gate to Section 3.1 in the revised paper._

---

> > ### Author Response · Authors · 2025-11-20
> > **Rebuttal Part Two**
> >
> > > W2. Because the augmented data are "reward-preserving state edits", it is unclear whether they correspond to valid trajectories under any real environment dynamics, potentially conflating causal variability with adversarial perturbation.
> >
> > > Q2. How does the method ensure that reward-preserving state perturbations remain within the true support of the environment’s dynamics? I do not think the proposed mitigation method is valid enough to guarantee this.
> >
> > We thank the reviewer for this rigorous question. We distinguish between _real dynamics consistency_ (which is unverifiable offline) and _state plausibility_ (which we strictly enforce), and we clarify why our method is the opposite of adversarial perturbation.
> >
> > **1. Verifying Real Dynamics Consistency is Infeasible in Offline RL**
> >
> > Offline RL benchmarks such as D4RL provide only observations, not the environment’s latent Markovian states. Following prior work (Decision Transformer, Decision ConvFormer), we construct “states” from observations. These states do not correspond to the true latent state, and the dataset does not provide ground-truth dynamics tuples $(s\_t, a\_t, s\_{t+1}, r\_t)$.
> >
> > Because the true environment state is unknown and no interaction is allowed, it is not feasible to learn or verify reliable transition dynamics in strictly offline RL. In preliminary experiments, we attempted to train a transition model, but consistent with known issues in offline dynamics learning, the learned dynamics were unstable and degraded overall performance. Under this constraint, enforcing  "true support of the environment’s dynamics" or "validating under any real environment dynamics" is infeasible.
> >
> > **2. We Enforce State Plausibility (Section 3.1 and Appendix H)**
> >
> > Instead, our method enforces the practical form of consistency that is achievable: state plausibility. We achieve this through two specific mechanisms:
> >
> > **(a) Move Band**
> >
> > Generation is restricted to a small, normalized neighborhood around the factual state (see
> >  Eq. (3)).
> > The Move histograms (Figures 5a and 6a in Appendix I.2) show that accepted edits lie in a narrow, controlled displacement band around $s\_t$.
> >
> > **(b) Hard Acceptance Gate**
> >
> > Every generated state must satisfy two independent constraints:
> >
> > * **Reward consistency:**
> >
> >   A counterfactual is accepted only if the CRM predicts a reward close to the observed reward $|\hat{f\_r}(s\_c, a\_t, z\_t) - r\_t| \le \epsilon\_r$.
> >
> > * **Trust-region constraint:**
> >
> >   Even if reward consistency holds, the edit is rejected unless $\||(s _c-s _t)/\sigma _s\|| _2\le\rho _{high}$.
> >
> > Empirically, the _acceptance heatmaps_ (Figures 5b and 6b in Appendix I.2) show structured regions of rejected samples, indicating that counterfactuals violating either constraint are systematically filtered out. The PCA overlays (Figures 5c and 6c in Appendix I.2) provide direct empirical evidence that accepted counterfactuals remain within the support of the observed data.
> >
> > _For clarity, we have moved specific details regarding the move band and acceptance gate to Section 3.1 in the revised paper._
> >
> > **3. The Model is Not Trained on Invalid Transitions**
> >
> > We believe the concern arises from interpreting $s\_c$ as creating a new transition $(s\_c, a\_t, s\_{t+1})$. In our method, this never occurs. The Decision Transformer does not model transition dynamics $P(s\_{t+1}\mid s\_t, a\_t)$; it treats states as conditioning tokens for action prediction. When a counterfactual state $s\_c$ replaces $s\_t$, the model learns from the valid tuple $(s\_c, a\_t, r\_t)$, but it is not trained to interpret $s\_{t+1}$ as the successor of $s\_c$. As a result, the model is never exposed to or required to validate a synthetic transition, and no inconsistency is introduced.
> >
> > _For clarity, we have added the related clarification in updated Section 3.1 (Line 222-225)_
> >
> > **4. Distinction from Adversarial Perturbation**
> >
> > We respectfully disagree that this conflates causal variability with adversarial perturbation. The two objectives are opposites:
> >
> > * Adversarial Perturbation: Seeks to _change_ the model's output (maximize error) using imperceptible noise that has _no_ causal effect on the ground truth.
> >
> > * Our Counterfactuals: Seek to _preserve_ the ground-truth reward (minimize error) using perceptible state edits ($s\_t \to s\_c$).
> >
> > By finding different states $s\_c$ that yield the same reward, we are not attacking the model; we are teaching it _invariance_. We force the policy to recognise that the specific pixel/feature values of the "distractors" do not change the value of the action, which is the definition of causal robustness. Our robustness experiments (Figure 3 & 4) show that DT and DC fail sharply when a spurious distractor is introduced, while UCF remains stable.
> >
> > _For clarity, we have summarized the counterfactual construction procedure in the updated Appendix H and moved specific details regarding the move band and acceptance gate to Section 3.1 in the revised paper._

---

> > > ### Author Response · Authors · 2025-11-20
> > > **Rebuttal Part Three**
> > >
> > > > W3. Any approximation error in the causal reward model directly affects the plausibility of generated counterfactual states. How to deal with this issue?
> > >
> > > We thank the reviewer for raising this important point. Approximation error is unavoidable in any learned component, so our method is designed to explicitly contain and limit the effect of CRM error rather than rely on perfect reward modeling.
> > >
> > > Our framework includes three mechanisms that limit the effect of CRM approximation error, supported by empirical analysis in _Appendix I.2_.
> > >
> > > **1. The CRM is a Local Consistency Gate, Not a Dynamics Model**
> > >
> > > The CRM predicts a one-step reward given $(s, a)$ and is never used to generate rollouts or transitions. Unlike model-based offline RL, errors do not accumulate over time. The policy is not exposed to synthetic next states, which prevents compounding model error.
> > >
> > > Our counterfactual generator uses a strict _generate-and-filter_ process (in Section 3.1 and Appendix H):
> > >
> > > **(a) Move Band**
> > >
> > > Generation is restricted to a small, normalized neighborhood around the factual state (see Eq. (3)).
> > > The _Move histograms_ (Figures 5a and 6a in Appendix I.2) show that accepted edits lie in a narrow, controlled displacement band around $s\_t$.
> > >
> > > **(b) Hard Acceptance Gate**
> > >
> > > Every generated state must satisfy two independent constraints:
> > >
> > > * **Reward consistency:**
> > >
> > >   A counterfactual is accepted only if the CRM predicts a reward close to the observed reward $|\hat{f\_r}(s\_c, a\_t, z\_t) - r\_t| \le \epsilon\_r$.
> > >
> > > * **Trust-region constraint:**
> > >
> > >   Even if reward consistency holds, the edit is rejected unless $\||(s _c-s _t)/\sigma _s\|| _2\le\rho _{high}$.
> > >
> > > Empirically, the _acceptance heatmaps_ (Figures 5b and 6b in Appendix I.2) show structured regions of rejected samples, indicating that counterfactuals violating either constraint are systematically filtered out. The _PCA overlays_ (Figures 5c and 6c in Appendix I.2) provide direct empirical evidence that accepted counterfactuals remain within the support of the observed data.
> > >
> > > _For clarity, we have moved specific details regarding the move band and acceptance gate to Section 3.1 in the revised paper._
> > >
> > > **3. KL Regularization**
> > >
> > > The CRM is implemented as a CVAE trained with an ELBO objective that includes a _KL regularization term_:
> > >
> > > $\text{KL}\big(q\_\phi(z \mid s, a, r)\\,\\|\\,p(z)\big)$,
> > >
> > > weighted by $\beta$. This KL regularization smooths the latent space and discourages degenerate or overly sharp posteriors, making the latent disturbance $z$ robust to small variations in $(s, a, r)$. In practice, this prevents the model from encoding noise or outliers as extreme latent codes and improves the stability of reward predictions used by the acceptance gate.
> > >
> > > **4. The Policy Learns from Factual Rewards**
> > >
> > > The policy is trained on $(s\_c, a\_t, r\_t)$, where $r\_t$ is the _factual reward_ from the dataset. The CRM is never used to label data for policy optimization. As a result, CRM error does not propagate into the training signal, and the augmentation process only modifies the conditioning states, not the reward label.

---

> > > > ### Author Response · Authors · 2025-11-20
> > > > **Rebuttal Part Four**
> > > >
> > > > > W4. The causal supervision and augmentation scheme remain untested in more complex or high-dimensional domains where the Markov structure or causal graph is less explicit.
> > > >
> > > > > Q5. In real-world domains where the causal structure is unknown or high-dimensional, how would the authors obtain or approximate the causal masks required for attention supervision?
> > > >
> > > > > Q6. Could the causal supervision be learned automatically or relaxed into a differentiable causal-discovery process?
> > > >
> > > > > Q7. How sensitive is the model’s performance to mis-specified causal masks or incomplete parent sets?
> > > >
> > > > We thank the reviewer for these related questions about causal masks and scalability. We address them together as they all concern the validity of the causal mask.
> > > >
> > > > **1. The Causal Mask is Universal, Not Arbitrary (Re: W4, Q5, Q7)**
> > > >
> > > > The mask is not a hand-crafted heuristic or a domain-specific graph. It encodes the _universal causal structure of a Goal-Conditioned Markov Decision Process (MDP)_.
> > > >
> > > > As detailed in _Section 2.2_ (referencing Peters et al., 2017; Zhang et al., 2020; Bennett et al., 2021; Shi et al.,
> > > > 2022 in Line 142) , the action $a\_t$ is causally determined by the current state $s\_t$ in standard RL. In the Decision Transformer context, the model generate actions based on future desired returns (return-to-go $\hat{G}\_t$).
> > > > Therefore, the parents of $a\_t$ are _definitionally_ $s\_t$ and $\hat{G}\_t$. This structural dependency is invariant across _all_ RL domains for DT-based methods, regardless of the dimensionality or complexity of the state space. We do not need to "approximate" a new mask for each environment; the MDP definition remains constant.
> > > >
> > > >
> > > > **2. We Have Evaluated on a Real-World Domain: Recommendation.  (Re: W4 & Q5)**
> > > >
> > > > We evaluate UCF on _nine D4RL datasets_ with different complexity levels. D4RL does not provide true Markovian states but only have observations. Following prior work such as Decision Transformer and Decision ConvFormer, we construct states directly from observations, which makes the setting more challenging due to irrelevant or partial information.
> > > >
> > > > To prove the generalization and robustness of our method, we also evaluate on a complex _real-world domain: recommendation_. Our experiments cover KuaiRand, KuaiRec, and VirtualTB, all of which are widely used, _real-world datasets_ for offline RL-based recommendation. Even in such settings:
> > > >
> > > > * The causal parent set of $a\_t$ remains unchanged.
> > > >
> > > > * The mask applies cleanly, because it encodes the _structural_ relationship, not any particular semantic interpretation of state features.
> > > >
> > > > Thus, scalability is not limited by the dimensionality of the state space. As long as the domain follows the standard RL formulation where actions depend on the current state (and RTG for DT-based models), the causal mask applies directly.
> > > >
> > > > **3. No risk of mis-specified or incomplete parent sets (Re: Q7).**
> > > >
> > > > Because the mask encodes the fundamental MDP definition, there is no risk of a "mis-specified" parent set in the standard standard RL. This causal structure is fixed across all RL environments for DT-based methods, including high-dimensional ones. Furthermore, the mask is implemented as a _soft loss_ ($\mathcal{L}\_{mask}$), not a hard constraint. The action token is _encouraged_ to attend to its causal parents, but the model retains the flexibility to attend to other context if necessary.
> > > >
> > > > **4.\. Automated Discovery vs. Soft Supervision (Re: Q6)**
> > > >
> > > > In our setting, learning the causal mask automatically is unnecessary. The parent set of $a\_t$ is fixed by the MDP structural equation as explained above. Automatically rediscovering this mask would add extra complexity without providing new information, because the discovered relations would still be between states, actions, rewards, and RTGs, which are already known from the MDP definition.
> > > >
> > > > There is a different research direction where causal discovery is meaningful: learning the causal relationships among the latent factors inside $s\_t$. Several recent works explore disentangling state features and identifying causal relations among them (Pitis
> > > > et al., 2022; Urpí al., 2024). But as stated in the introduction, such methods rely on strong structural assumptions, such as
> > > > factored environments with independent entities that are not applicable to all environments. Instead, our approach does not assume such structure, making it broadly applicable to standard continuous-control benchmarks such as Mujoco.
> > > >
> > > > Furthermore, our _soft supervision_ provides flexibility. The scalar $\lambda$ controls the strength of the causal prior. Tuning such environment-dependent scalars is standard practice in RL. Our ablation (Figure 2) shows that $\lambda$ adapts naturally to environmental complexity without altering the underlying causal mask.

---

> > > > > ### Author Response · Authors · 2025-11-20
> > > > > **Rebuttal Part Five**
> > > > >
> > > > > > W5. From the experimental results (Tables 1,2&3), it seems that the performance of the proposed method does not show obvious advantage compared to the baselines, with a bit improvement only.
> > > > >
> > > > > We thank the reviewer for their assessment, but we respectfully disagree with the conclusion that the performance gains are marginal. The empirical advantage of UCF is _not_ measured solely by moderate improvements on clean data, but by its consistent improvements across domains and, more importantly, its robustness under spurious correlations.
> > > > >
> > > > > Across all domains (9 datasets in MuJoCo and 3 datasets in recommendation), our method shows _consistent gains_ on different sequential decision-making problems (Table 1 and Table 2), not isolated spikes. In offline RL, stability and consistency are regarded as strong indicators of method quality.
> > > > >
> > > > > More importantly, our approach also aims to _educe spurious feature reliance_ and _improve causal robustness_. This is clearly shown in our robustness experiment (Figure 3), where baseline models such as DT and DC show a dramatic collapse in performance when a deterministic spurious distractor is introduced, while UCF maintains stable performance.
> > > > >
> > > > > Following the suggestion of another reviewer, we also added a second experiment using a _stochastic distractor_ (Updated to Robustness to Stochastic Distractors with Figure 4 in Appendix G.4). The distractor is weakened by random flips, but it still induces a substantial degradation in DT and DC while UCF remains stable. This demonstrates that our method’s advantages are not tied to the severity of the correlation but arise from recovering the reward-relevant structure.
> > > > >
> > > > > > Q4 Could the proposed counterfactual augmentation be reframed as a form of invariance or robustness regularization rather than genuine counterfactual reasoning?
> > > > >
> > > > > We thank the reviewer for this thoughtful question. While the learned policy shows invariance to spurious features, this is not a regularization effect but a consequence of applying SCM-grounded counterfactual reasoning.
> > > > >
> > > > > As explained above (also in Section 3 and updated Appendix H), our method is a faithful, step-by-step implementation of _Pearl’s three-stage process for counterfactual reasoning_. This distinguishes our method from invariance-based regularizers.
> > > > > Invariance regularizers do not ensure that edited samples correspond to valid causal alternatives; they only push representations to be stable under generic noise.
> > > > > Our procedure instead uses _structural constraints_ during generation: the edited state must be reward-preserving and must remain inside the empirical state manifold. Thus, the model is trained on counterfactual tuples $(s\_c, a\_t, r\_t)$ that correspond to meaningful, causally equivalent alternatives, rather than arbitrary or adversarial perturbations.
> > > > >
> > > > > The empirical evidence further supports this interpretation. The PCA overlays and acceptance heatmaps (Appendix I.2) show that accepted counterfactuals remain well inside the data manifold and maintain reward consistency. Crucially, the variation between $s\_t$ and $s\_c$ is concentrated in the distractor dimensions (Figures 3 and 4), confirming that the generator modifies the spurious factors while preserving the causal ones.

---

> ### Author Response · Authors · 2025-11-27
>
> Dear Reviewer D5mJ,
>
> As the discussion period is approaching its close, we wanted to kindly follow up and check whether there are any remaining questions or points that you would like us to clarify regarding our rebuttal.
>
> We hope our responses have adequately addressed your concerns, but please let us know if any points require further clarification. We would value the opportunity to engage with you again before the deadline.
>
> Thank you for your time.

---

### Official Review · Reviewer_okqb · 2025-11-04

**Soundness:** 4
**Presentation:** 3
**Contribution:** 3
**Rating:** 6
**Confidence:** 3

**Summary:**

This paper tackles the problem of "causal confusion" in Transformer-based offline reinforcement learning, where models learn to exploit spurious correlations in a fixed dataset instead of the true causal mechanisms of the environment. The authors propose the Unified Causal Transformer (**UCF**), a framework that addresses this issue from both a data and a model perspective.

The key contributions are:

1. **Counterfactual Data Augmentation**: A novel pipeline for data augmentation in a strictly offline setting. It first trains a Causal Reward Model (CRM) using a CVAE to abduce latent exogenous factors from transitions. It then uses a Counterfactual State Generator (CSG) to produce new, reward-preserving states by editing existing states while keeping the action and inferred exogenous factors constant. This exposes the model to causal variations not present in the observational data.

2. **Causally-Structured Hybrid Architecture**: A hybrid model that combines modality-specific 1D convolutions to capture local Markovian dynamics with a final global self-attention layer for policy reasoning.

3. **Causal Attention Supervision**: The global attention mechanism is explicitly supervised with a causal mask, forcing attention heads to focus only on the direct causal parents of an action (the current state and return-to-go), thereby instilling a strong and correct inductive bias.


The authors validate UCF on standard D4RL locomotion and Antmaze navigation tasks, as well as on sequential recommendation datasets. The results show consistent performance gains over state-of-the-art baselines and demonstrate significantly improved robustness against engineered spurious correlations.

**Strengths:**

1.  Novel and Principled Framework: The core idea of jointly enforcing causal consistency at both the data level (via counterfactuals) and the model level (via architectural priors and supervision) is elegant, powerful, and novel. It provides a holistic solution to the well-known problem of causal confusion.

2. Robust Offline Counterfactual Generation: The proposed mechanism for generating reward-preserving counterfactuals (CRM + CSG) is well-designed for the strictly offline setting. The inclusion of a "move band" to ensure edits are neither trivial nor off-manifold is a thoughtful and critical detail that enhances the quality of the augmentation.

3. Direct and Effective Inductive Bias: The causally supervised attention mechanism is a very direct and effective way to bake the known causal structure of an MDP into a Transformer. This is a much stronger and more targeted inductive bias than simply using convolutions for locality.

4. Ablation clarity: The study isolates the contributions of counterfactual augmentation, architectural causal supervision, and mask weight λ, providing clear insights.

**Weaknesses:**

1. **Dependence on CRM accuracy (i.e., accurate reward modeling)**: The success of the counterfactual augmentation hinges on the fidelity of the CRM; inaccuracies can propagate to poor or unrealistic augmentations.

2. **Evaluation domains**: Despite covering robotics and recommendation, all tasks rely on relatively standard benchmarks; evaluation on complex partially observable or high-dimensional real-world domains would further strengthen claims.

3. **No explicit causal discovery**: The causal mask structure is predefined (state $\rightarrow$  action, RTG $\rightarrow$ action). The method does not learn or adapt causal graphs dynamically, which may limit scalability.

4. **Fixed Causal Graph for Supervision** (building on [2+3] above): The supervised attention relies on a predefined causal mask based on the known MDP structure (i.e., $a_t$ depends on $s_t$ and $G_t$). While effective, this may be less straightforward to apply in settings where the causal structure is unknown, more complex, or only partially observable (e.g., in POMDPs).

**Questions:**

1. How sensitive is UCF’s performance to inaccuracies in the causal mask (e.g., if an incorrect parent set is imposed)? Could the mask be learned or adapted via causal discovery methods?

2. CRM Robustness: What mechanisms prevent the CVAE-based Causal Reward Model from overfitting to dataset noise or producing implausible latent factors? Was any regularization or early stopping based on held-out validation used?

3. Did you observe cases where the augmentation harmed performance (e.g., under poor reward modeling or narrow data coverage)? If so, how were these mitigated?

4. The counterfactual generation pipeline is a key component. Could you provide some insight into the acceptance rate of the generated states? How does the choice of the move band (p_low, p_high) and the reward tolerance affect both the acceptance rate and the diversity of the final augmented dataset?

### Minor Typos

- Line 446: ref. missing for ACE

---

> ### Author Response · Authors · 2025-11-20
> **Rebuttal Part One**
>
> > W1. **Dependence on CRM accuracy (i.e., accurate reward modeling)**: The success of the counterfactual augmentation hinges on the fidelity of the CRM; inaccuracies can propagate to poor or unrealistic augmentations.
>
> > Q2. CRM Robustness: What mechanisms prevent the CVAE-based Causal Reward Model from overfitting to dataset noise or producing implausible latent factors? Was any regularization or early stopping based on held-out validation used?
>
> We thank the reviewer for raising the concern that the counterfactual augmentation depends on the accuracy of the CRM, and we address both W1 and Q2 together since they concern the same underlying issue.
>
> **1. The CRM is a Local Consistency Gate, Not a Dynamics Model**
>
> The CRM predicts only a one-step reward given $(s, a)$ and a latent context $z$. Crucially, it is _never used to generate rollouts or synthetic next states or reward_. Unlike model-based offline RL, where errors compound over time, the CRM's error is local and cannot lead to exposure to compounding model error during policy evaluation. Its role is solely to act as a consistency checker for the current time step.
>
> **2. Two Safeguards Prevent Inaccurate CRM Predictions from Producing Bad Counterfactuals**
>
> Our counterfactual generator uses a strict _generate-and-filter_ process (in Section 3.1 and Appendix H):
>
> **(a) Move Band**
>
> Generation is restricted to a small, normalized neighborhood around the factual state (see
>  Eq. (3)). The Move histograms (Figures 5a and 6a in Appendix I.2) show that accepted edits lie in a narrow, controlled displacement band around $s\_t$.
>
> **(b) Hard Acceptance Gate**
>
> Every generated state must satisfy two independent constraints:
>
> * **Reward consistency:**
>
>   A counterfactual is accepted only if the CRM predicts a reward close to the observed reward $|\hat{f\_r}(s\_c, a\_t, z\_t) - r\_t| \le \epsilon\_r$.
>
> * **Trust-region constraint:**
>
> Even if reward consistency holds, the edit is rejected unless $\||(s _c-s _t)/\sigma _s\|| _2\le\rho _{high}$.
>
> Empirically, the _acceptance heatmaps_ (Figures 5b and 6b in Appendix I.2) show structured regions of rejected samples, indicating that counterfactuals violating either constraint are systematically filtered out. The _PCA overlays_ (Figures 5c and 6c in Appendix I.2) provide direct empirical evidence that accepted counterfactuals remain within the support of the observed data.
>
> **3. KL Regularization**
>
> The CRM is implemented as a CVAE trained with an ELBO objective that includes a _KL regularization term_:
>
> $\text{KL}\big(q\_\phi(z \mid s, a, r)\\,\\|\\,p(z)\big)$,
>
> weighted by $\beta$. This KL regularization smooths the latent space and discourages degenerate or overly sharp posteriors, making the latent disturbance $z$ robust to small variations in $(s, a, r)$. In practice, this prevents the model from encoding noise or outliers as extreme latent codes and improves the stability of reward predictions used by the acceptance gate.
>
> _For clarity, we have summarized the counterfactual construction procedure in the updated Appendix H and moved specific details regarding the move band and acceptance gate to Section 3.1 in the revised paper._
>
> > W2. **Evaluation domains**: Despite covering robotics and recommendation, all tasks rely on relatively standard benchmarks; evaluation on complex partially observable or high-dimensional real-world domains would further strengthen claims.
>
> We thank the reviewer for this comment. We would like to clarify that our experiments already cover both _standard offline RL benchmarks_ and a _real-world domain_.
>
> **1. D4RL is the widely used benchmark for offline RL.**
>
> We evaluate UCF on _nine D4RL datasets_ with different complexity levels. D4RL does not provide true Markovian states but only have observations. Following prior work such as Decision Transformer and Decision ConvFormer, we construct states directly from observations, which makes the setting more challenging due to irrelevant or partial information.
>
> **2. We also evaluate on a real-world domain: recommendation.**
>
> To prove the generalization and robustness of our method, we also evaluate on a complex _real-world domain: recommendation_. Our experiments cover KuaiRand, KuaiRec, and VirtualTB, all of which are widely used, real-world datasets for offline RL-based recommendation.
>
> **3. The main goal of UCF is causal robustness, not only improved performance.**
>
> Across both robotics and recommendation, our method gives stable improvements, but its core benefit is made clear in the _robustness experiments_. With a _deterministic distractor_ (Figure 3), baseline models such as DT and DC suffer a major drop in performance, while UCF stays stable. Following another reviewer’s suggestion, we also added a _stochastic distractor_ (Updated to Appendix G.4 Robustness to Stochastic Distractor with Figure 4). Even when the distractor is weakened by random flips, baseline models still degrade noticeably while UCF remains strong.

---

> > ### Author Response · Authors · 2025-11-20
> > **Rebuttal Part Two**
> >
> > > W3. **No explicit causal discovery**: The causal mask structure is predefined (state $→$ action, RTG $→$ action). The method does not learn or adapt causal graphs dynamically, which may limit scalability.
> >
> > > W4. **Fixed Causal Graph for Supervision** (building on \[2+3] above): The supervised attention relies on a predefined causal mask based on the known MDP structure (i.e., $a\_t$ depends on  $s\_t$ and $G\_t$). While effective, this may be less straightforward to apply in settings where the causal structure is unknown, more complex, or only partially observable (e.g., in POMDPs).
> >
> > > Q1. How sensitive is UCF’s performance to inaccuracies in the causal mask (e.g., if an incorrect parent set is imposed)? Could the mask be learned or adapted via causal discovery methods?
> >
> >
> > We appreciate these related comments  about causal masks and address them together.
> >
> >
> > **1. The Causal Mask is Universal, Not Arbitrary (Re: W3, W4, Q1)**
> >
> > The mask is not a hand-crafted heuristic or a domain-specific graph. It encodes the _universal causal structure of a Goal-Conditioned Markov Decision Process (MDP)_.
> >
> > As detailed in _Section 2.2_ (referencing Peters et al., 2017; Zhang et al., 2020; Bennett et al., 2021; Shi et al.,
> > 2022 in Line 142) , the action $a\_t$ is causally determined by the current state $s\_t$ in standard RL. In the Decision Transformer context, the model generate actions based on future desired returns (return-to-go $\hat{G}\_t$).
> > Therefore, the parents of $a\_t$ are _definitionally_ $s\_t$ and $\hat{G}\_t$. This structural dependency is invariant across _all_ RL domains for DT-based methods, regardless of the dimensionality or complexity of the state space. We do not need to "approximate" a new mask for each environment; the MDP definition remains constant.
> >
> > **2. MDP and POMDP (Re: W4)**
> >
> > Our method is developed for _MDP-based offline RL_, which is the standard setting for Decision Transformers.
> >
> > Even in a POMDP, the structural causal relationship between the underlying state $s _t$, the action $a _t$ and the reward $r _t$ does not change. The $s _t$ is still the parent of $a _t$ (Huang et al., 2022).
> > The DT architecture still conditions on $(s\_t, \hat{G}\_t)$, so the same parent set applies regardless of whether the environment originated from a POMDP or MDP.
> >
> > Thus, the causal mask is fully aligned with the structural assumptions used by all DT-based methods.
> >
> > _Huang, Biwei, et al. "Action-sufficient state representation learning for control with structural constraints." International Conference on Machine Learning. PMLR, 2022._
> >
> > **3. No Performance Sensitivity to “Incorrect” Masks (Re: Q1)**
> >
> > Since the mask encodes the fixed MDP causal structure, the question of incorrect parent sets does not arise in standard RL settings. Additionally, the mask is implemented as a _soft loss_, not a hard constraint. The action token is encouraged, but never forced to attend to its parents. This gives the model flexibility while still injecting helpful causal structure. Ablations in Figure 2 show that the model adapts naturally as $\lambda$ varies.
> >
> > **4. Automated Discovery vs. Soft Supervision (Re: W3 & Q1)**
> >
> > In our setting, learning the causal mask automatically is unnecessary. The parent set of $a\_t$ is fixed by the MDP structural equation as explained above. Automatically rediscovering this mask would add extra complexity without providing new information, because the discovered relations would still between states, actions, rewards, and RTGs, which are already known from the MDP definition.
> >
> > There is a different research direction where causal discovery is meaningful: learning the causal relationships among the latent factors inside $s\_t$. Several recent works explore disentangling state features and identifying causal relations among them (Pitis et al., 2022; Urpí al., 2024). But as stated in the introduction, such methods rely on strong structural assumptions, such as factored environments with independent entities that are not applicable to all environments. Instead, our approach does not assume such structure, making it broadly applicable to standard continuous-control benchmarks such as Mujoco.
> >
> > Furthermore, our _soft supervision_ provides flexibility. The scalar $\lambda$ controls the strength of the causal prior. Tuning such environment-dependent parameters is standard practice in RL. Our ablation (Figure 2) shows that the model can adapt to environmental complexity without altering the underlying causal mask.

---

> > > ### Author Response · Authors · 2025-11-20
> > > **Rebuttal Part Three**
> > >
> > > > Q3. Did you observe cases where the augmentation harmed performance (e.g., under poor reward modeling or narrow data coverage)? If so, how were these mitigated?`
> > >
> > > > Q4. The counterfactual generation pipeline is a key component. Could you provide some insight into the acceptance rate of the generated states? How does the choice of the move band (p\_low, p\_high) and the reward tolerance affect both the acceptance rate and the diversity of the final augmented dataset?
> > >
> > > We thank the reviewer for these related questions. We address them together because they concern the same mechanism: the _generate-and-filter_ counterfactual generation module.
> > >
> > > **1. We did not See Augmentation Harmed Performance in our Experiments (Re: Q3)**
> > >
> > > Across all 12 environments (9 D4RL + 3 recommendation), we did not observe any performance degradation due to augmentation. As explained above, our counterfactual generator uses a strict _generate-and-filter_ process:
> > >
> > > **(a) Move Band**
> > >
> > > Generation is restricted to a small, normalized neighborhood around the factual state (see
> > >  Eq. (3)).
> > >
> > > **(b) Hard Acceptance Gate**
> > >
> > > Every generated state must satisfy two independent constraints:
> > >
> > > * **Reward consistency:**
> > >
> > >   A counterfactual is accepted only if the CRM predicts a reward close to the observed reward $|\hat{f\_r}(s\_c, a\_t, z\_t) - r\_t| \le \epsilon\_r$.
> > >
> > > * **Trust-region constraint:**
> > >
> > >   Even if reward consistency holds, the edit is rejected unless $\||(s _c-s _t)/\sigma _s\|| _2\le\rho _{high}$.
> > >
> > > **2. Empirical Evidence: Ablation Studies on Counterfactual Generation in Appendix I  (Re: Q3 &Q4):**
> > >
> > > Appendix I includes a dedicated ablation study on counterfactual generation. It provides:
> > >
> > > * Ablations on the reward gate
> > >
> > > * Ablations on the band constraint
> > >
> > > * Explaination of the choices of move band $(\rho _{low},\rho _{high})$ through three aspect diagnostics: Move histograms, Acceptance heatmaps and PCA overlays
> > >
> > > **For Q3 (harmful augmentation):**
> > >
> > > * The PCA overlays (Figures 5c and 6c) show that accepted samples lie on the real data manifold.
> > >
> > > * The acceptance heatmaps (Figures 5b and 6b) show that low-confidence CRM regions are rejected.
> > >
> > > * The ablation on removing the reward gate shows that harmful samples appear only when the gate is removed, confirming that the gate is doing the intended job.
> > >
> > > **For Q4 (acceptance rate, move band and reward tolerance):**
> > >
> > > * The move histograms (Figures 5a and 6a) show that accepted states fall in a narrow, calibrated displacement region, justifying our choice of the move band of $(0.2,0.5)$.
> > >
> > > * The acceptance heatmaps (Figures 5b and 6b) shows that our chosen range $(0.2,0.5)$ for maintains high acceptance (around 90\%~ 95\% on Hopper-medium and around 70\%~75\% for HalfCheetah-medium) while keeping edits within a safe move region. Tightening the lower bound (e.g., $\rho _{low}=0.3$) sharply decreases acceptance, discarding many otherwise valid counterfactuals. Conversely, relaxing the upper bound (e.g., $\rho _{high}>0.6$ ) increases acceptance but allows off-manifold edits (because the $\rho _{high}$ term is used in Acceptance Gate).
> > >
> > > * The PCA overlays (Figures 5c and 6c) show that accepted samples remain on-manifold.
> > >
> > > * As stated in Appendix I.1, $\tau\_r$ is not treated as an arbitrary hyperparameter. We automatically set
> > > $\varepsilon\_r = 0.1 \times \mathrm{std}(r),$
> > > which scales the tolerance relative to the intrinsic reward variability of each dataset. This provides a stable and data-dependent way to calibrate the confidence threshold of the CRM. Table 9 further shows that removing the reward-consistency gate harms performance, confirming that this constraint is necessary for safe counterfactual augmentation.
> > > Because the acceptance criterion includes
> > > $|\hat{f\_r}(s\_c, a\_t, z\_t) - r\_t| \le \epsilon\_r$,
> > > decreasing $\tau\_r$ makes the reward-consistency condition harder to satisfy. As a result, the acceptance rate decreases and the final augmented dataset becomes smaller.

---

> ### Author Response · Authors · 2025-11-27
>
> Dear Reviewer okqb,
>
> As the discussion period is approaching its close, we wanted to kindly follow up and check whether there are any remaining questions or points that you would like us to clarify regarding our rebuttal.
>
> We hope our responses have adequately addressed your concerns, but please let us know if any points require further clarification. We would value the opportunity to engage with you again before the deadline.
>
> Thank you for your time.

---

### Meta-Review · Area_Chair_4BsJ · 2026-01-10

**Summary:**

Overall, the suggested rating of the paper is borderline, leaning toward the paper is not yet ready for publication, mainly because reviewers raised concerns about experimental validation and positioning, and these concerns were not clearly addressed during discussion.

Experimental validation / strength of gains: Reviewers D5mJ and m5nA noted that the improvements appear small / not clearly decisive vs. baselines, and requested stronger evidence that UCF provides clear benefit beyond standard DT-style modeling. While the paper emphasizes robustness/causal advantages, reviewers felt the empirical results were not yet sufficiently compelling to fully support the strength of the claims (D5mJ, m5nA).

Counterfactual validity / positioning of augmentation: Reviewers questioned whether the proposed state-edit counterfactuals correspond to meaningful causal alternatives or are closer to invariance / robustness regularization, especially since transition-consistency cannot be validated in strict offline RL. This creates uncertainty around whether the method’s framing as causal counterfactual reasoning is fully justified (D5mJ, m5nA).

Conceptual ambiguity affecting confidence: Reviewer L8qK raised the concern that edited states may imply invalid/impossible transitions, and expressed doubts about the conceptual correctness and the causal attention supervision component, which lowers overall confidence in correctness.

Positive note on rebuttal: The authors made a strong effort in the rebuttal to clarify offline RL assumptions, explain why validating true dynamics is infeasible, provide detailed justification for the causal mask (as soft supervision), and add extra robustness experiments (including stochastic distractors) along with extensive diagnostics (acceptance heatmaps / PCA overlays / histograms). These clarifications are helpful and demonstrate serious engagement. However, the core concerns about positioning and empirical conclusiveness remain partially unresolved.

The paper has also been discussed in the AC panel including the area chair and the senior area chair, concluding the decision.

**Reviewer Concerns:**

Reviewer okqb

Addressed: CRM robustness + causal mask misunderstanding

Outstanding: None major (already borderline-positive)

Reviewer D5mJ

Addressed: why state interventions (offline constraint), plausibility gates/diagnostics, mask ≠ Markov restriction, robustness framing

Outstanding: may still dislike state edits conceptually (dynamics unverifiable offline)

Reviewer m5nA


Addressed: no identifiability claim, dynamics-model infeasibility, added stochastic distractor exp

Outstanding: may still worry about “impossible triplets” / transition validity interpretation

Reviewer L8qK

Addressed: exogenous variable misunderstanding, “disentangled” term fixed, mask clarified as soft/limited

Outstanding: main blocker remains: counterfactual state seen as implying invalid transitions

**Reviewer Scores:**

Unfortunately, the reviewers have not participated in the discussion period.

---

### Decision · Program_Chairs · 2026-01-26

Reject